# Vibrationally-dependent molecular dynamics in mutual neutralisation reactions of molecular oxygen ions

Mathias Poline[1], Arnaud Dochain[1,2], Stefan Rosén[1], MingChao Ji [1], Henrik Cederquist [1], Henning Zettergren [1], Henning T. Schmidt [1], Mats Larsson[1], Shaun G. Ard[3], Nicholas S. Shuman[3], Albert A. Viggiano[3] & Richard D. Thomas [1] ✉

Product distributions and dynamics of low-collision-energy mutual neutralisation reactions involving even simple molecular ions are largely unknown. Reactions which involve oxygen ions, e.g., $O_2^+$ with $O^-$, are expected to be important in atmospheric phenomena such as sprites and in high-pressure air or oxygen discharges. Here we show, by combining cryogenically stored-and-merged ion beams with coincident product-imaging techniques, that the $O_2^+$ with $O^-$ mutual neutralisation reaction results predominantly in dissociation of the $O_2^+$ molecule. Three competing reaction pathways yields both $O(^3P)$ (84%) and $O(^1D)$ (16%) products, but no $O(^1S)$ products. Analysis of the momentum-correlated dynamics of the reaction reveals the dominance of two-step mechanisms involving the $3p\lambda_u$ and $3s\sigma_g$ Rydberg states of $O_2$. Furthermore, use of the $^{16,18}O_2^+$ isotopologue shows that the reaction products strongly depend on the vibrational levels of the $O_2^+$ ion for the channel leading to two $O(^1D)$ products.

The evolution of plasma environments is defined and governed by intricate balances between ionising processes, chemical rearrangements, and neutralisation reactions such as mutual neutralisation (MN) and dissociative recombination (DR). Measuring and explaining these processes in detail is fundamental to understanding and modelling non-local thermal equilibrium (non-LTE) environments, from low-temperature interstellar media[1–3] through cool atmospheric plasma[4–8] and stellar atmospheres[9–11], to high-pressure discharges and high-temperature plasma[12,13].

The $O_2^+ + O^-$ mutual neutralisation reaction is expected to occur in essentially any high-pressure air or oxygen discharges[14]. Both ions are present in the mesosphere, and, although $O^-$ is nominally a minor constituent, the $O_2^+ + O^-$ MN reaction is expected to be of significance, given the large cross sections of MN reactions. For example, recent ionospheric models[15] suggest that the related reaction of $O^+$ with $O^{-7,16}$ is a significant source of the $O(^5S^o) \rightarrow O(^3P)$ 135.6 nm airglow emission, as observed by the Hopkins Ultraviolet Telescope[17]. Another natural phenomenon where the MN of $O_2^+$ with $O^-$ is expected to be important is sprites (see e.g.,[18,19]), which are upwards discharges occurring in conjunction with lightning. The primary sprite activity occurs for only a few tens of microseconds, but certain features persist for much longer. Because they are transient in a non-LTE environment, the full atmospheric ionic reaction scheme does not have time to develop, and the charged-particle density can reach values as high as $\approx 10^{11}$ m$^{-3}$ [19]. Therefore, MN, including MN between $O_2^+$ and $O^-$, is expected to be important in such environments.

In the MN of $O_2^+$ with $O^-$ at very low collision energies in the centre-of-mass frame ($E_{c.m} \approx 0$ eV), the following final sets of products

[1]Department of Physics, Stockholm University, Stockholm SE-10691, Sweden. [2]Institute of Condensed Matter and Nanosciences, Université Catholique de Louvain, Louvain-la-Neuve B-1348, Belgium. [3]Space Vehicles Directorate, Air Force Research Laboratory, Kirtland AFB, Albuquerque 87117 New Mexico, USA. ✉e-mail: rdt@fysik.su.se

are energetically allowed:

$$O_2^+(^2\Pi_g) + O^-(^2P) \rightarrow O_2^*(^{2S+1}\Lambda) + O(^3P) + 1 - 10\,\text{eV} \quad (1)$$

$$\rightarrow O(^3P) + O(^3P) + O(^3P) + 5.45\,\text{eV} \quad (2)$$

$$\rightarrow O(^1D) + O(^3P) + O(^3P) + 3.50\,\text{eV} \quad (3)$$

$$\rightarrow O(^1D) + O(^1D) + O(^3P) + 1.53\,\text{eV} \quad (4)$$

$$\rightarrow O(^1S) + O(^3P) + O(^3P) + 1.28\,\text{eV} \quad (5)$$

where the quoted energies correspond to the kinetic energy released in each reaction outcome, given that both reactants are in their electronic and rovibrational ground states. The quoted values are determined using the NIST atomic spectra database[20] and the NIST Chemistry WebBook[21]. For $E_{c.m} \approx 0$ eV, reaction (1) represents all possible final states of pure MN with no molecular dissociation, while reactions (2–5) represent every possible final channel for MN accompanied by molecular dissociation.

Until recently, experimental studies of MN involving molecular ions in flow tubes[22–24] and merged-beams experiments[25,26] were limited to measurements of overall reactivities without detailed information on the reaction mechanisms or the final states formed. The Double ElectroStatic Ion Ring ExPeriment (DESIREE) facility[27,28], with its combination of stored and merged ion beams and coincident imaging detection, has now made such studies possible[8,29]. The properties of MN reaction products, including their kinetic energies, are of great importance in oxygen-containing plasmas. There, products with large kinetic energies may excite other plasma constituents in collisions or contribute to plasma cooling when leaving the region. In addition, electronically excited atomic and molecular products may instead emit characteristic red (630 nm), green (558 nm)[30,31], and blue photons[32] responsible for atmospheric airglows.

A recent study of the MN of NO$^+$ and O$^-$ at DESIREE reported that MN is predominantly accompanied by dissociation of the NO molecule[8], which contrasts with earlier flowing afterglow results for the MN between NO$^+$ and a number of anions[33–35]. In addition, it was concluded that the reaction occurred via a two-step process involving a pre-dissociating NO$^*$ Rydberg state[8]. However, very little is known about the role of the rovibrational energy in molecular ions on the MN dynamics. To study this effect, long-time ion-beam storage is extremely useful, as it allows the ions to relax vibrationally. The reaction can be studied at both short and long storage times, and the results can be compared. In the related DR process between a free electron and O$_2^+$, a strong dependence is observed on the vibrational quanta, with the quantum yields of different oxygen product states varying up to a factor of 2 as a function of vibrational quantum number[30,31,36,37]. Using DESIREE's unique capabilities, we are now able to investigate the influence of vibrational excitations in a molecular MN reaction for the first time. Thus, a good understanding of the properties of atomic and molecular ions and the details of their interactions in natural plasma, including mutual neutralisation between O$_2^+$ with O$^-$, will help to better understand atmospheric phenomena.

Here, in the most detailed study so far on the MN of O$_2^+$ with O$^-$ at low collision energies, $E_{c.m} \leq 0.1$ eV, we report that the reaction is completely dominated by dissociation into three products, in which the process proceeds in a two-step mechanism via Rydberg states in O$_2$, which we identify. In addition, we find that the intensity of one of the reaction channels depends strongly on the vibrational state of the O$_2^+$ parent ion.

## Results

General details about the two ion-beam storage rings, data acquisition, subsequent analysis, and Monte Carlo simulations of the reaction outcome, are found in the Methods section, the Supplementary Information (SI), and in references[6,8,27,28]. Briefly: the two ion beams are stored in their separate rings, merged into the interaction region where MN may take place, and then de-merged into their respective rings. The kinetic energy released in the reaction is manifested as recoil momenta of the neutral particles relative to the centre of mass of the collision system. The recoil momenta, and thus the kinetic energy of each particle, can then be directly measured in the lab frame. Coincidence measurements allow the number of products to be determined, and the total kinetic energy released into the products after each MN event, $E_{K_f}$, to be measured. This energy corresponds to the sum of: (i) the energy released in the reaction, $E_K$, given in reactions (1–5) for zero ion-ion collision energy with O$_2^+$ and O$^-$ in their electronic ground states; (ii) the collision energy $E_{c.m.}$ of the two ions; (iii) the change in rovibrational internal energy between the reactants and products, $\Delta E(v,J)$, i.e., $E_{K_f} = E_K + E_{c.m.} - \Delta E(v,J)$.

### Kinetic energy release spectra

Analysis of the coincidence events allows separate kinetic-energy release spectra to be obtained for the two-body and three-body data[8], which, in general, is expected to correspond to excitation and fragmentation of O$_2$ formed in the MN process, respectively. The contribution of the three-body events to the two-body spectrum is inferred by randomly selecting two out of three events from the three-body data. After comparison of the two spectra (see the SI), we conclude that no measurable two-body channels are populated, and that the data are completely dominated by three-body MN events. We thus conclude that the oxygen molecule predominantly fragments when O$_2^+$ and O$^-$ are mutually neutralised in low-energy collision, $E_{c.m} \leq 0.1$ eV.

The three-body $E_{K_f}$ spectrum obtained from analysis of all coincidence events at a collision energy of $E_{c.m} \leq 0.1$ eV is shown in Fig. 1a. Experimental data (filled circles) are plotted with statistical uncertainties. The upper $x$-axis shows the equivalent total 3D atom displacement, TD, on the detector (see the SI). Data are consistent with the reaction proceeding via channels (2–4) only. The O($^3P$)+O($^3P$)+O($^3P$) channel (2) and the O($^1D$)+O($^3P$)+O($^3P$) channel (3) are found to be the dominant products, and approximately equally important, while the O($^1D$)+O($^1D$)+O($^3P$) channel (4) is much lower in intensity. Although the kinetic energy released into channel (5) lies only 0.3 eV lower in energy than that released into channel (4), and could be attributed to the tiny feature observed at lower $E_{K_f}$ values (labelled with a "*" in Fig. 1a), we show later that this assignment can be ruled out based on the reaction dynamics. Given the small separation (TD -2 cm), we assign this tiny feature to false coincidences (see the "Methods" section and the SI for more details).

Given the ion source conditions, see e.g.,[8], and based on the observed shift and broadening of the peaks, the rovibrational energy, $E_{v,J_{avg}}$, of the O$_2^+$ ions is described by an initial 3000 ± 500 K Boltzmann distribution. While this accurately describes the fit to the two dominant peaks (as seen in the $v$-numbered bars in Fig. 1a), the smaller peak is only described by non-Boltzmann, individually-fit high-$v$ contributions. This strongly indicates that this reaction depends on the vibrational state of the O$_2^+$ ions.

To investigate this further, the experiment was repeated using the $^{16,18}$O$_2^+$ isotopologue, which introduces a small permanent molecular dipole moment, allowing IR-vibrational cooling during ion storage[8]. The vibrational lifetimes have been calculated previously, and are on the order of seconds[38]. A storage lifetime of 60 seconds was chosen, which ensured close to complete depletion of the higher ($v \geq 4$) vibrational states, without requiring unnecessarily long total data acquisition times (see the SI). The resulting spectrum is shown in

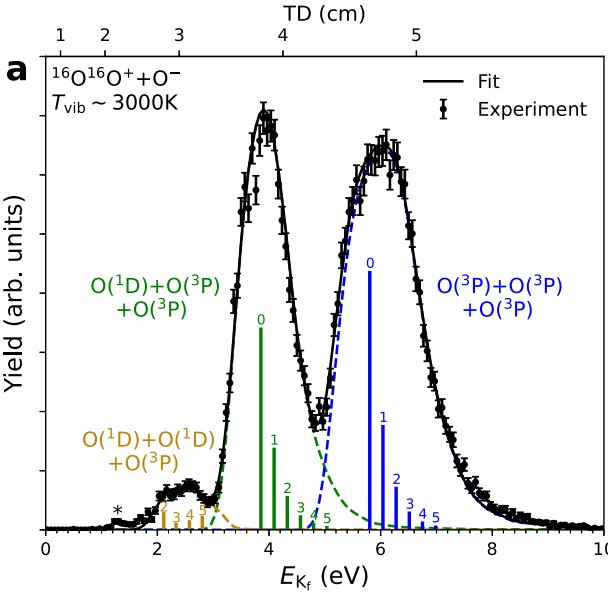

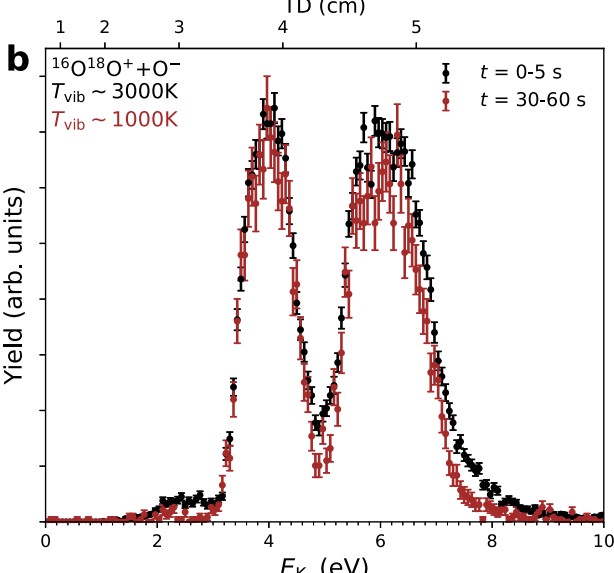

**Fig. 1 | Coincident three-body total kinetic energy release, $E_{K_f}$, and corresponding Total Displacement, *TD*, distributions.** Experimental data are plotted as filled circles, and the error bars are the standard deviations. Individually simulated distributions are plotted as dashed lines, and the solid line represents the total fit, with the positions of the $O_2^+$ ($v, J_{avg}$) rovibrational energies represented by the vertical bars. **a** Data for MN of $^{16,16}O_2^+ + O^-$ at storage times 0-10 s (vibrational temperature $T_{vib}$ ~ 3000 K). The feature labelled with a '*' is discussed in the main text. **b** Data for MN of $^{16,18}O_2^+ + O^-$ at storage times 0–5 s (black circles, $T_{vib}$ ~ 3000 K) and 30–60 s (red circles, $T_{vib}$ ~ 1000 K).

**Table 1 | Population and quantum yields of the different sets of oxygen atoms in the MN of $O_2^+$ with $O^-$**

| Channel | 3000 K | 1000 K [a] |
|---|---|---|
| O($^3$P)+O($^3$P)+O($^3$P) | 56.0 ± 0.8% | 54.8 ± 1.0% |
| O($^1$D)+O($^3$P)+O($^3$P) | 40.4 ± 0.7% | 44.8 ± 0.9% |
| O($^1$D)+O($^1$D)+O($^3$P) | 3.6 ± 0.7% | 0.4 ± 0.3% |
| O($^3$P) yield | 84.1 ± 1.1% | 84.8 ± 1.3 % |
| O($^1$D) yield | 15.9 ± 0.7% | 15.2 ± 0.6 % |

Data are listed for $O_2^+$ effective vibrational temperatures of 3000 K and 1000 K.
[a] Obtained using the $^{16,18}O_2^+$ isotopologue.

Fig. 1b, which displays $E_{K_f}$ data from MN events analysed from two different slices of ion-storage times.

As expected, the features get narrower with time, due to the vibrational cooling of $^{16,18}O_2^+$, see e.g.,[8]. As the lifetimes differ largely from higher to lower vibrational states, a thermal distribution is not expected after storage for more than 30 s. However, the later data is reasonably well described by a $T_{vib} = 1000$ K vibrational distribution (see the SI). When compared to the $^{16,16}O_2^+$ data, the first small peak appears significantly smaller, likely due to an isotope effect, but more importantly it almost vanishes during cooling. This indicates that only the higher vibrational states of the $O_2^+$ ion contribute to this channel. We determine the population of the different sets of electronically excited oxygen atoms at the two corresponding vibrational temperatures: namely, 3000 K and 1000 K, with an uncertainty of ± 500 K, as shown in Table 1.

The quantum yield is found to be dominated by ground state O($^3$P) with about 84%. In comparison, the DR $O_2^+$ produces about 50% excited atoms, including a small yield of O($^1$S), thus contributing to the green and red airglows[31,36]. Here, however, due to the relatively lower yield of O($^1$D), MN is thereby not expected to be a major source of airglow, but rather results mainly in ground-state O($^3$P) fragments with high kinetic energies. The quantum yield varies little between the two temperatures: the loss of O($^1$D) atoms from the third channel at the

lower temperature is partly compensated by an increase in the second. Therefore, the quantum yield is expected to be nearly the same for any vibrational temperature.

## Mutual neutralisation reaction dynamics

The three-body reaction dynamics can be studied by converting particle position and time data into Dalitz coordinates, $\eta_1$, $\eta_2$ (see the "Methods" section, the SI, and refs. [8,39,40]):

$$\eta_1 = \frac{E_1 - E_2}{\sqrt{3}E}; \quad \eta_2 = \frac{2E_3 - E_2 - E_1}{3E} \quad (6)$$

In Fig. 2a(i), we indicate the relations between the velocity vectors of three separating atoms of equal mass at different Dalitz coordinates, representing different distributions of the total kinetic energy release $E_{K_f} = E = E_1 + E_2 + E_3$, among three atoms with individual kinetic energies $E_1$, $E_2$, and $E_3$, respectively. The axes for $E_1$, $E_2$, and $E_3$ are indicated by the dotted lines in Fig. 2a(i). In Dalitz representations, knots and diagonal lines reflect the intrinsic dynamics which led to the atoms receiving their observed kinetic energies and momenta, as indicated by the break-up geometries shown in Fig. 2a(i). Since the three oxygen atoms are identical, each Dalitz plot has three-fold symmetry.

Dalitz distributions for each of the three peaks observed in the experimental $E_{K_f}$ data from Fig. 1a are plotted in Fig. 2a(ii-iv), respectively. The dominating three-fold diagonals in Fig. 2a(ii) indicate directly that the mechanism forming these products is a two-step process. In the first step, the outermost (most loosely bound) electron in $O^-$ is transferred to $O_2^+$ forming an excited neutral molecule $O_2^*$, which dissociates later in a second step, well separated in time from the first step. That is, the intermediate products $O_2^*$ and O are spatially well separated when $O_2^*$ dissociates. The time-scale for the dissociation must be considerably shorter than the flight-time of the products from the interaction point to the detector, which, in the current experiment, is ≈ 10 μs.

The observed features of high intensities in Fig. 2a(ii) correspond to a well-defined fraction of the maximum available energy, $E_{K_f}$, for

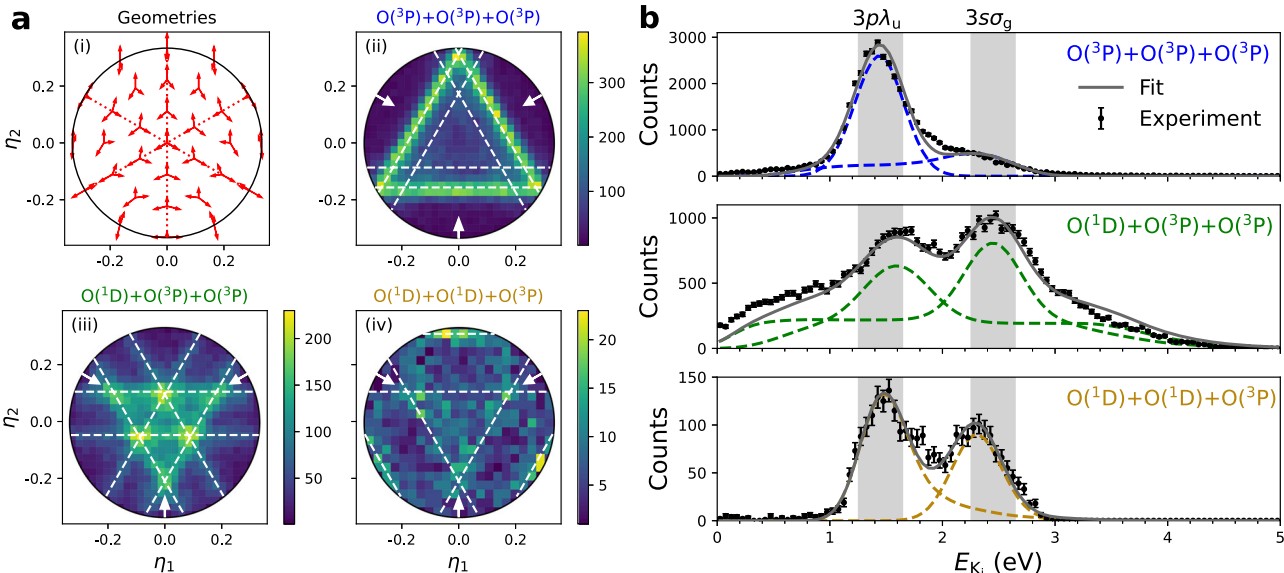

**Fig. 2 | Dalitz representation analysis for the observed three-body product channels with extracted intermediate kinetic energy release $E_{K_i}$ distributions.** **a** Dalitz representation analysis showing (i) the relation between Dalitz coordinates $\eta_1$, $\eta_2$ and break-up geometries, with the red dotted lines indicating the energy axis of each atomic product; (ii), (iii), and (iv): Experimental Dalitz plots for the $E_{K_f}$ in the following reaction channels: (2) O($^3$P)+O($^3$P)+O($^3$P); (3) O($^1$D)+O($^3$P)+O($^3$P); and (4) O($^1$D)+O($^1$D)+O($^3$P), respectively. The white arrows denote the energy axis of each oxygen atom, and the white dashed lines are shown to highlight the observed structures. **b** Extracted $E_{K_i}$ distributions for the formation of the different inter-mediate states of O$_2^*$ in the following reaction channels: (2) O($^3$P)+O($^3$P)+O($^3$P); (3) O($^1$D)+O($^3$P)+O($^3$P); and (4) O($^1$D)+O($^1$D)+O($^3$P), respectively. Experimental data are plotted as filled circles, and the error bars are the standard deviations. Individually simulated distributions are plotted as dashed lines, and the solid line represents the total fit. The $E_{K_i}$ regions indicated by the grey vertical bands correspond to electronically excited states in O$_2$ associated with two electronic configurations ($3p\lambda_u$ ($\lambda = \Sigma$, $\Pi$, $\Delta$) and $3s\sigma_g$).

one of the O atoms. This unambiguously shows that the intermediate O$_2^*$ state is formed with a specific amount of internal energy in the initial electron transfer step of the reaction. Some additional events are observed inside the dominating triangle of the Dalitz plot in Fig. 2a(ii), although of much lower intensity, indicating a second intermediate state. White dashed lines are shown to highlight these features. Although Fig. 2a(iii) shows different structures to that of Fig. 2a(ii), the features again indicate a two-step process. Here, however, two very distinct sets of lines are clearly seen, whose intersections give rise to the bright knots, meaning that this product channel is definitely reached via two different intermediate states. Finally, though with fewer statistics, structures are also identified in Fig. 2a(iv) (as high-lighted by the white dashed lines) for the production of O($^1$D)+O($^1$D)+O($^3$P).

Given the two-step nature of these reactions, further dynamics can be retrieved, and the reaction can be described as follows:

$$O_2^+ + O^- \rightarrow O_2^* + O + E_{K_i} \rightarrow O + O + O + E_{K_f}, \quad (7)$$

i.e, momentum conservation allows us to determine the kinetic-energy release, $E_{K_i}$, related to the intermediate first step (see the "Methods" setion and the SI). Here, we assume that the O atom no longer participates in the reaction after the initial electron-transfer step, i.e. no further energy is then exchanged between O and O$_2^*$. Calculated $E_{K_i}$ data are plotted in Fig. 2b, with the top, middle, and bottom panels corresponding to the intermediate states in O$_2$ involved in the reactions producing O($^3$P)+O($^3$P)+O($^3$P), O($^1$D)+O($^3$P)+O($^3$P), and O($^1$D)+O($^1$D)+O($^3$P), respectively. In the case of reactions producing O($^1$D)+O($^3$P)+O($^3$P), no definitive selection could be made, and a broad background is present. However, the clear peaks observed correspond directly to the features in the associated Dalitz plot. The peaks are narrower in these figures than in the three-body $E_{K_f}$ data, indicating that the intermediate neutral state possesses a similar equilibrium geometry to the parent cation, such that vertical electron capture processes are favoured. For the fully ground-state products, O($^3$P)

+O($^3$P)+O($^3$P) (top panel), an intense peak is observed, which reinforces the conclusion that one intermediate state is predominantly involved. However, the broad shoulder at larger $E_{K_i}$ does confirm a contribution from a second intermediate state. In the middle panel, corresponding to production of O($^1$D)+O($^3$P)+O($^3$P), the two distinct peaks, with similar amplitude (green curve, corresponding to Fig. 2a(iii)), similarly confirm the involvement of two different intermediate states. Finally, in the bottom panel, corresponding to production of O($^1$D)+O($^1$D)+O($^3$P), despite the lower level of statistics, two peaks appear in the gold curve, corresponding to Fig. 2a(iv), at similar positions but with different relative amplitudes.

## Discussion

The present results are explained in the framework of the two-step reaction model[8]. The $E_{K_i}$ regions indicated by the grey vertical bands in Fig. 2b correspond to electronically excited states in O$_2$ associated with two electronic configurations, namely the $3p\lambda_u$ ($\lambda = \Sigma$, $\Pi$, $\Delta$) and $3s\sigma_g$ configurations. As a $p$-electron is transferred from O$^-$, formation of $\Sigma$-, $\Pi$-, and $\Delta$-states is possible. In Fig. 3a, b, we plot the potential-energy curves (adapted from van der Zande et al.[41], and Morrill et al.[42]) of these two configurations and with these state symmetries. The O$_2^+$ potential has been shifted by the electron affinity (EA) of O such that the energy of the intermediate O$_2^*$ Rydberg states directly corresponds to the intermediate kinetic energy release $E_{K_i}$, and the energies of the different final O+O+O states to $E_{K_f}$. Contributions from rotational energies are neglected.

The peak close to 2.5 eV in Fig. 2b, ascribed to the $3s\sigma_g$ config-uration, matches well the expected $E_{K_i}$ of the C $^{1,3}\Pi_g$ Rydberg state, corresponding to the black potential-energy curve in Fig. 3a. The predissociation pathways of this state are well known from measure-ments of electron capture by O$_2^+$ from neutral Cs by van der Zande et al.[41]. They reveal that this state primarily dissociates diabatically to the O($^1$D)+O($^3$P) channel, in agreement with the present results, and this occurs via the dissociative 1 $^3\Pi_g$ state (the green curve in Fig. 3a), which has a favourable crossing with the C state at $v = 1$. Dissociation to the

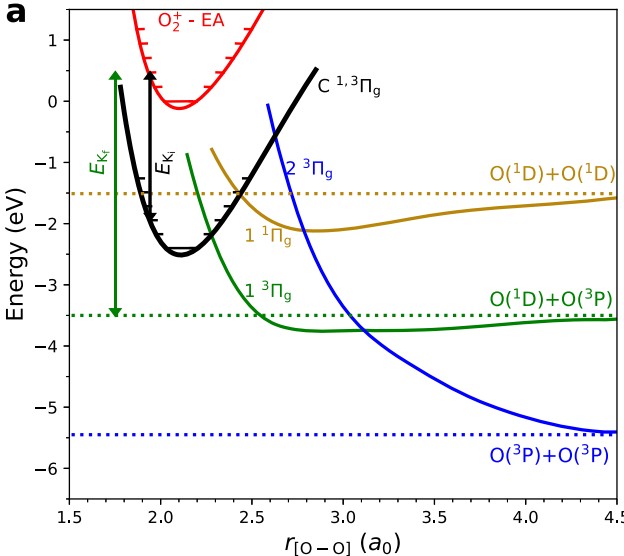

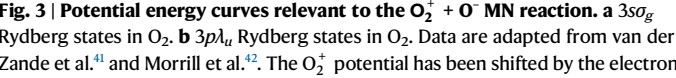

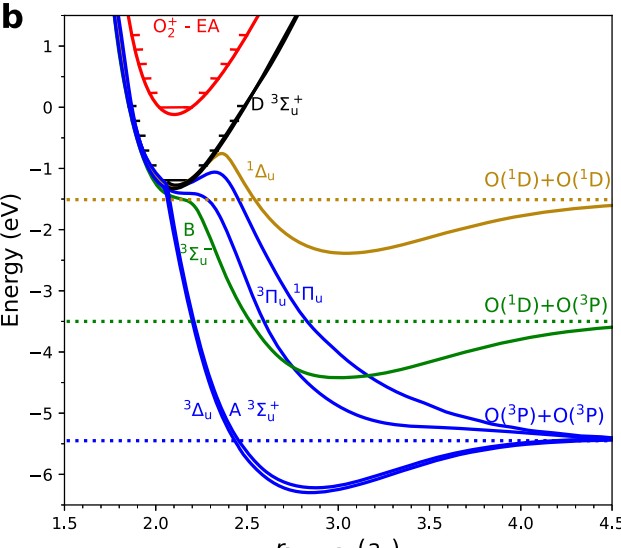

**Fig. 3 | Potential energy curves relevant to the $O_2^+ + O^-$ MN reaction. a** $3s\sigma_g$ Rydberg states in $O_2$. **b** $3p\lambda_u$ Rydberg states in $O_2$. Data are adapted from van der Zande et al.[41] and Morrill et al.[42]. The $O_2^+$ potential has been shifted by the electron

affinity (EA) of O to illustrate the kinetic energies $E_{K_i}$ and $E_{K_f}$ in the two-step process for the reaction $O_2^+(v=2) + O^- \rightarrow O_2^*(v=2) + O(^3P) + E_{K_i} \rightarrow O(^1D) + O(^3P) + O(^3P) + E_{K_f}$, as an example.

$O(^3P)+O(^3P)$ channel is also open but with a much weaker coupling[41], which explains the smaller amplitude of this channel in the $O(^3P)+O(^3P)$ $+O(^3P)$ data (top panel in Fig. 2b). For higher vibrational levels, van der Zande et al. also report on dissociation to the $O(^1D)+O(^1D)$ channel, due to the favourable crossing with the $1\,^1\Pi_g$ state (the gold curve in Fig. 3a)[41], which is consistent with our observation of a peak for this channel at an intermediate kinetic energy release around $E_{K_i} = 1.5$ eV (the gold curve in Fig. 2b, bottom panel).

Contributions from $3p\lambda_u$ states are more difficult to disentangle, as several states with allowed $\Sigma-$, $\Pi-$, and $\Delta-$symmetries lie very close in energy. In this region, strong mixing between valence and Rydberg states is reported[43]. For example, the $^3\Pi_u$ state, which plays an important role in the DR of $O_2^+$, has its Rydberg and valence states only differing by one orbital, resulting in strong interaction. The $^3\Sigma_u^-$ state, responsible for the Schumann-Runge continuum in the photoabsorption spectrum of $O_2$, also exhibits strong mixing with Rydberg states above 9 eV[44] (here corresponding to $E_{K_i} < 1.7$ eV). We therefore rely on more advanced theoretical calculations to get a better understanding of these dissociation pathways.

From $O_2$-electron impact studies[45], and theoretical calculations by Lewis et al.[43,46] and Morrill et al.[42], we find that most of the dissociating states shown in Fig. 3b can be described adiabatically as quasi-bound states. Only one state directly couples to the $O(^1D)+O(^3P)$ channel, namely the B $^3\Sigma_u^-$ state (the green curve in Fig. 3b), whereas dissociation to the $O(^3P)+O(^3P)$ pair can occur via a number of states (the blue curves in Fig. 3b). Since these states lie close in energy, and are described by the same valence electron, the electron coupling mainly depend on the involved state symmetries. For the vibrational coupling, since the electronic states possess a vibrational wavefunction similar to that of the $O_2^+$ ion, vertical electron capture processes are expected to dominate. However, perturbations due to the strong interaction with the valence states may occur. This could explain the observed slight shift to higher energy of this peak in the $O(^1D)+O(^3P)$ data (middle panel in Fig. 2b), which may be attributed to capture into the flat region of the B $^3\Sigma_u^-$ potential.

Intriguingly, coupling to the $O(^1D)+O(^1D)$ channel is also found to involve a small energy barrier, corresponding to $v=2$ in the $^1\Delta_u$ state (the gold curve in Fig. 3b). Assuming vertical electron capture, the observed $O_2^+$-vibrational dependence of this channel has a clear explanation: Only vibrationally excited $O_2^+(v \geq 2)$ would overcome this

energy barrier, and these states cool away in $^{16,18}O_2^+$ during ion-storage as shown in Fig. 1. The final outcome of the lower vibrational levels of the $^1\Delta_u$ state is, however, less clear. A recent experimental study has identified radiative transitions from this state to the a $^1\Delta_g$ state[47]. Such stabilised $O_2$ products are not observed in our data (see the SI), which suggests predissociation via other states is favoured, possibly via rotational coupling.

The exclusion of the $O(^3P)+O(^1S)$ channel, i.e., reaction (5), can be understood from this model: there are no readily accessible potentials that couple out to this channel from the Rydberg states involved here. Dissociative recombination studies indicate that this exit channel is only directly accessible from $O_2^+(v > 9)$[31,48], as confirmed by the theoretical calculations by Guberman[49]. It is therefore no surprise that it is not observed in the present experiment.

In order to evaluate the contribution from each intermediate state, Monte Carlo simulations were performed in the framework of the Free-Rotor model. This assumes that the $O_2$ molecule rotates freely, with no exchange of energy between the two neutral products, $O_2$ and O, after electron transfer or during dissociation, see e.g.,[8,50]. The result of the fits from the simulated distributions is shown in Fig. 2b, and is used to determine the branching between the two intermediate $O_2$ states. The retrieved values are shown in Table 2. The quoted uncertainties reflect the model limitations, as discrepancies are observed between the best fit and the experimental data. This is noticeable in the fit to the data leading to the production of $O(^3P)$ $+O(^3P)+O(^3P)$ (top panel in Fig. 2b), where two shoulders arise on both sides of the main peak, which can not be solely explained by the contribution from the $3s\sigma_g$ state. In a similar fashion, in the fit to the $O(^1D)+O(^3P)+O(^3P)$ data (middle panel in Fig. 2b), the left and right sides of the peaks are found to be underestimated/overestimated, respectively. These shoulders arise mainly due to the misidentification of the atomic oxygen product in the electron-transfer step, which may not be well described in our model, although we cannot rule out the possibility of non-vertical electron capture processes contributing to these features. However, in all three cases, the main features of the peaks are well reproduced by the model, which supports the idea that vertical transitions into these Rydberg states dominate the spectra.

By combining the results of Tables 1, 2, the total branching into the two intermediate classes of states is found to be 43% and 57% for the $3s\sigma_g$ and $3p\lambda_u$ states, respectively (for $T_{vib} = 3000$ K). These results

should be most useful for benchmarking theoretical models of mutual neutralisation reactions involving molecules.

In summary, extracting product information and reaction dynamics in the mutual neutralisation of molecular ions is a challenge. Combining the already powerful techniques of merged ion beams and coincident product imaging with cryogenic ion storage, as is uniquely offered by the DESIREE facility, provides not only high-quality data but a crucial window into the molecular dynamics, allowing us to disentangle the detailed mechanisms behind the mutual neutralisation process, and to identify specific intermediate states. Here, this has been applied to the mutual neutralisation of $O_2^+$ and $O^-$. We find that the reaction predominantly results in complete dissociation into three different sets of electronic states of atomic oxygen products. Through analysis of the energy partitioning, we conclude that the reaction occurs through a sequential process. From the available potential-energy curves, we determine that the intermediate step involves two classes of $O_2$ Rydberg states, which we identify, and where strong Rydberg-valence interactions control the dynamics. Storage of the $^{16,18}O_2^+$ isotopologue allows us to investigate the effects of vibrational cooling, which reveal a strong dependence of the reaction outcome on the initial $O_2^+$ vibrational population for one of the channels, and that this effect is explained by an energy barrier associated with the intermediate state coupling out to this exit channel. As all of these aspects have not been reported before, these results may be expected to be important for the development of models of high-pressure air and oxygen discharges, including transient atmospheric phenomena such as sprites.

**Table 2 | Branching fractions into the intermediate states leading to the final three sets of atomic products**

| Final state | Intermediate state | Branching fraction [%] |
|---|---|---|
| $O(^3P)+O(^3P)+O(^3P)$ | $3p\lambda_u$ | $68 \pm 14$ |
| | $3s\sigma_g$ | $32 \pm 9$ |
| $O(^1D)+O(^3P)+O(^3P)$ | $3p\lambda_u$ | $42 \pm 9$ |
| | $3s\sigma_g$ | $58 \pm 9$ |
| $O(^1(^1D)+O(^1D)+O(^3P)$ | $3p\lambda_u$ | $60 \pm 11$ |
| | $3s\sigma_g$ | $40 \pm 11$ |

## Methods

The experiment was carried out at the double electrostatic ion-beam storage ring DESIREE at Stockholm University, presented in Fig. 4[27,28]. The setup consists of two electrostatic ion-beam storage rings, each of 8.7 m circumference, with a common straight section including a set of seven, independently controllable drift tubes. Molecular $O_2$ was used as a source gas to produce the $O_2^+$ cation beam in an electron cyclotron resonance (ECR) ion source (Pantechnik Monogan). The $O_2^+$ ions were extracted and accelerated to a final beam energy of 17.504 keV, and then injected into the asymmetric ring, as shown in Fig. 4. The $O^-$ beam, produced by caesium sputtering of a $SnO_2$ cathode (National Electrostatics Corp.), was extracted and accelerated to 7.504 keV and injected into the symmetric ring.

Two sets of experiments were undertaken under the same experimental conditions: in the first set, $^{16,16}O_2^+$ ions were produced and stored for about 10 seconds; in the second set, the mixed $^{16,18}O_2^+$ isotopologue was used and stored for 60 s. In both cases, this ensures that the ions de-excite from any electronically excited state, and, in the latter case, it makes it possible to study the effects of vibrational cooling. For the $^{16,18}O_2^+$ isotopologue, vibrational lifetimes are on the order of a few seconds[38]. Rotational cooling lifetimes are several orders of magnitude longer (see e.g.,[51,52]), and the $J-$state distribution of the ions is not expected to change significantly from what they had at their production, and is considered constant.

During each orbit, the $O_2^+$ and $O^-$ ion-beams are overlapped in a common straight section of drift-tubes, before being de-merged back into their respective rings. A voltage of +840 V was set on three of the tubes to accelerate and decelerate the anions and cations, respectively, in order to match their velocities and reach the lowest collision energy possible. At DESIREE, this is typically between 50 and 100 meV, due to the small angular spreads of the beams. This is satisfactory, in relation to the atmospheric conditions of interest (between 300 and 1000 K, corresponding to a collision energy of 30–130 meV). High collision energy events from the unbiased region are easily separated out by their large time separations.

The neutral particle detection apparatus, located at about 1.77 metres from the centre of the selected interaction region, consists of a triple stack of 75 mm-diameter microchannel plates (MCPs) with a similarly sized phosphor-screen anode. Light from the phosphor is

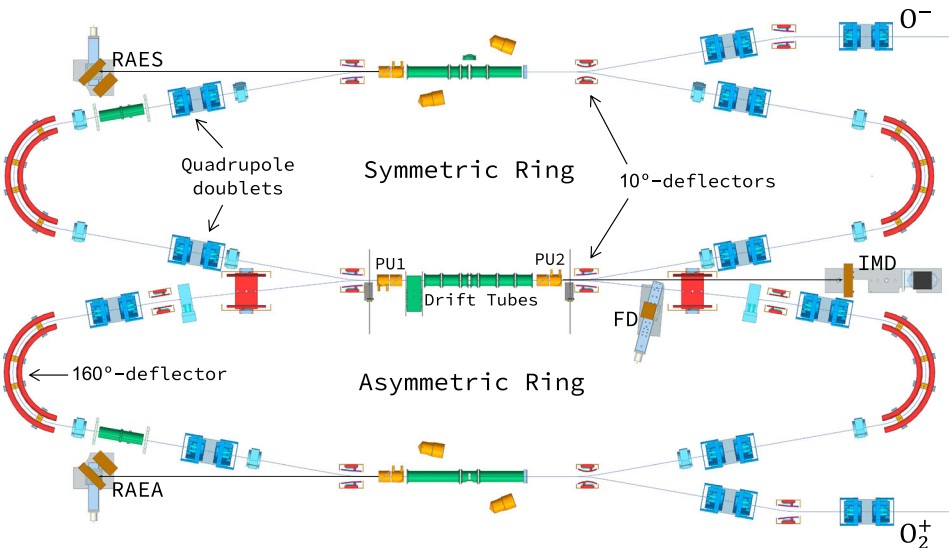

**Fig. 4 | Schematic of the experimental setup at DESIREE.** The two oppositely charged ion beams are injected and stored in their respective rings by the use of quadrupoles and deflectors. Between the two pick-up (PU) electrodes, they are merged together and interact freely. The neutrals formed in the MN reactions continue straight to the imaging detector (IMD). Resistive-anode encoder-based detectors are used for neutral beam monitoring (RAES, RAEA) and charged-fragment detection (FD).

coupled out from the cryogenic environment, and imaged onto an optical camera with a $256 \times 256$ pixel array (TPX3CAM, see e.g.,[8,53,54]). Each pixel in the camera is effectively its own photomultiplier (PMT) system, allowing for the time information of each event to be recorded. The positions of each neutral are retrieved from the centre of the illuminated pixels on the camera.

For an MN event taking place at a distance $L$ from a detector, with ions moving at speed $v$ in the laboratory system, the kinetic energy of a product with mass $m_i$ in the centre-of-mass system is:

$$E_i = \frac{v^2}{2L^2} m_i \mathbf{r_i}^2, \tag{8}$$

where $\mathbf{r_i} = (\Delta x_i, \Delta y_i, v \Delta t_i)$ is the position vector of the particle, relative to the centre-of-mass of the products. Coincidence measurements allow us to determine the number of products, i,e.. $i = 2, 3$, and the sum of the kinetic energies of the products. This yields the final kinetic energy released in the reaction $E_{K_f}$, which is the sum of the reaction's kinetic energy release, $E_K$, the collision energy, $E_{c.m.}$, and the rovibrational internal energy of the cation, $E_{v, J}$. For a three-body break-up $E_{K_f} = E_1 + E_2 + E_3$, the intermediate kinetic energy in the reaction $E_{K_i}$ is retrieved by momentum conservation from the kinetic energy of the neutralised anion. In this case, it corresponds to the sum of the initial kinetic energy release $E_{K_i}$, the collision energy $E_{c.m.}$, and the difference in internal energy $\Delta E_{v, J}$, between the initial and intermediate state. The MN reaction can, in principle, lead to both two- and three-body products, and these are treated independently.

Analysis of all coincidence events, including effects such as centre-of-mass filtering[8] allows separate kinetic energy release spectra to be obtained for the two-particle and three-particle data. As described in Supplementary Note 1.1, evaluation of such data for the former scenario is straightforward (see, e.g.,[6,7]), and the resultant experimental $E_{K_f}$ spectrum (blue circles) is shown in Suppl. Fig. 1. However, three-body events, in which one particle is not detected but which still satisfies centre-of-mass filtering, thus also contribute to the two-body data. This contribution, which is evaluated using the three-body data (see Fig. 1a), is plotted (red circles) in Suppl. Fig. 1. The true, two-body $E_{K_f}$ spectra are determined from the comparison of these two data sets.

In an MN reaction, which produces three atomic products, energy and momentum conservation allows a wide range of energy sharing. Here, as described in Supplementary Note 1.2, analysis of all coincident events is based on the work of U. Müller, P. C. Cosby and co-workers[39,40], who developed a scheme to parameterise the kinetic energy and momentum given to three products from a fragmentation process, and, in the subsequent transformation into the laboratory frame, how these then would be measured by a suitable detector. Using this approach, the total displacement (TD) of the fragments and their kinetic energy $E_{K_f}$ are determined.

The dynamics in the fully dissociative neutralising reactions is investigated by converting the kinetic energy of the three fragments into a set of two energy-weighted coordinates, $\eta_1$, $\eta_2$, and then representing these data in an energy and momentum conserving Dalitz plot[8]. The dynamics of the reaction are evaluated based on where each event is located in the Dalitz plot, as this is determined both by the energy fraction each particle received and how they received that fraction. Evaluation of Dalitz representations is used to determine if the fully dissociative reactions occur instantaneously, e.g., as the electron is transferred, or in two distinct steps, e.g., the transfer and dissociation are sufficiently separated in time. Both processes give rise to different features in the Dalitz plot. As described in the Suppl. Note 1.4., the free-rotor model developed by Hishikawa and co-workers[50] is used to model a two-step reaction, where the kinetic

energy obtained by the two fragments in the initial electron transfer step is determined from analysis of the Dalitz plot.

To compare with experimental observations, three-body Monte Carlo simulations of the energy and momentum distribution of the atomic products in the framework of a sequential break-up of a free rotor are carried out. Simulated datasets are generated and then processed using an identical workflow as used for the experimental data. The simulated data are used to both reproduce the experimentally observed Dalitz distributions, where separate, independent simulations are performed for the different contributing intermediate states, as well as to reproduce the experimentally observed $E_{K_f}$ spectra. The simulated processed data are fit to the experimental data to extract, e.g., the intermediate and final state branching fractions.

## Data availability

All data used in generating the figures in the manuscript are deposited in the following Zenodo repository[55]. All data needed to evaluate the conclusions in the paper are present in the paper and/or the Supplementary Information.

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

## Acknowledgements

This work was performed at the Swedish National Infrastructure, DESIREE (Swedish Research Council Contract No. 2017-00621 and 2021-00155). The views expressed are those of the authors and do not reflect the official guidance or position of the Department of the Air Force, the Department of Defence (DoD), or the U.S. government. The appearance of external hyperlinks does not constitute endorsement by the United

States DoD of the linked websites, or the information, products, or services contained therein. The DoD does not exercise any editorial, security, or other control over the information you may find at these locations. The authors disclose support for the research of this work from the Air Force Office of Scientific Research under Award Nos. FA9550-19-1-7012; FA8655-24-1-7004 (R.D.T.) and under AFOSR-25RVCOR006 (S.G.A., N.S.S., A.A.V.), from the Swedish Research Council under contract numbers 2023-038333 (H.C.), 2022-02822 (H.T.S.), and 2020-03437 (H.Z.), and from the Knut and Alice Wallenberg Foundation under contract number 2018.0028 (R.D.T., H.C., H.Z., H.T.S.).

## Author contributions

Investigation: M.P., A.D., S.R., MC.J., H.C., H.Z., H.T.S., M.L., S.G.A., N.S.S., A.A.V. and R.D.T. Funding acquisition: H.C., H.Z., H.T.S., A.A.V. and R.D.T. Supervision: H.Z., H.T.S. and R.D.T. Writing - original draft: M.P. and R.D.T. Writing - review & editing: M.P., A.D., S.R., MC.J., H.C., H.Z., H.T.S., M.L., S.G.A., N.S.S., A.A.V. and R.D.T.

## Funding

## Competing interests

The authors declare no competing interests.
