## [Transparent Peer Review file · Nature Communications]

Vibrationally-dependent molecular dynamics in mutual neutralisation reactions of molecular oxygen ions

Corresponding Author: Dr Richard Thomas

Version 0:

Reviewer comments:

Reviewer #1

(Remarks to the Author)

In this paper, the authors observed the reaction products from the mutual neutralization of O_2^+ and O^- by combining the merged ion beams and coincident product imaging with the cryogenic ion storage ring DESIREE. They disentangled the detailed mechanisms behind the mutual neutralization process and identified specific intermediate states. They found that the reaction predominantly results in complete dissociation into one of three different sets of electronic states of three atomic oxygen products. They also concluded that the reaction occurs through a sequential process.

In the introduction, the motivation of this study and the background, especially connected with “sprites” as a natural phenomenon, were comprehensively described. The dissociation channels were disentangled by taking advantage of the imaging technique, and the involved dynamics were successfully clarified. As for the theoretical approach, the existing results and information were utilized fully. Thus, I thoroughly evaluated their achievement. However, before this paper, the authors have already published an outstanding paper dealing with $NO^+ O^-$ in Phys. Rev. Lett. [ref. 9]. The experimental technique and analysis procedure employed are the same or very similar. I suppose, partly from this point of view, they determined the title of this paper as “Disentangling vibrationally-dependent....”

In the first figure in this paper, namely Fig. 1(a), I found the spectra of $^{16}O^{16}O^+$ in (a) at the storage time of 0-10s. Without a permanent dipole moment, this molecule has no chance of being vibrationally cooled. Then, they showed the spectra of $^{16}O^{18}O^+$ in Fig. 1(b), which should be vibrationally cooled at the storage time of 30-60s. Finally, I found the spectra of $^{16}O^{18}O^+$ at the storage time of 30-60s in Fig. S2(b) in Supplementary Information. I feel they should use Fig. S2(b) as the first figure instead of the present Fig.1(a) or at least add Fig. S2(b) to Fig.1 in the main text.

As evidence of vibrationally-dependent radiative cooling, depression of the population of higher vibrational levels is expected, as shown in colored bars in Fig. 1. However, the observed peak structure is “not vibrationally resolved,” and the corresponding change experimentally observed was not so noticeable (although the higher energy tail was suppressed). But, I understand the most impressive observation is the profile in the range of 1.5-2.0 eV, which is attributed to the component of $O(1D)+O(1D)+O(3P)$. They are explained by the characteristic small potential barrier shown in Fig.3(b).

Here, from the viewpoint point of atmospheric chemistry, $^{16}O^{18}O^+$ must be a minor component, and $^{16}O^{16}O^+$ is not influenced by vibrational cooling.

Judging from these advantages and drawbacks, I consider that, without a doubt, the presented results are excellent; however, I hesitate to conclude that they are worthy of acceptance in Nat. Comm. I'd like to get a response from the authors and other referees' opinions.

Finally, I list one question in Supplementary Information
SI P.5 line 217 and line 221:

I found two different explanations for Fig. S2 (b). I guess the former is the experimental data, and the latter is the simulation result. I think the former is correct.

Reviewer #2

(Remarks to the Author)

This manuscript presents an experimental study on the mutual neutralisation reaction between O_2^+ and O^- at low collision energy, using DESIREE in combination with coincident product imaging. It was found that at low collision energy, this reaction proceeds predominantly via a two-step mechanism via two classes of Rydberg states of O_2 , resulting in three atomic products via three different reaction channels. The occurrence of one of these channels depends on the vibrational temperature of the O_2^+ . The data and observations could be interpreted with the help of available potential energy curves for O_2 , such that new insight is provided into the reaction mechanisms for this mutual neutralisation.

In previous research using similar methods, mutual neutralisation of H_3O^+ with OH^- , and NO^+ with O^- was investigated (refs. 21 and 8, respectively). For NO^+ with O^- , one three-body channel was present and it was found that the reaction occurred via a two-step process involving a pre-dissociating NO^* Rydberg state. Here, multiple three-body channels are open, their branching ratio is determined, and information about the intermediate states involved in the two-step process is obtained. So this is really new compared to what has been done for NO^+ with O^- .

It is challenging to investigate mutual neutralisation reactions involving molecular ions, and this study shows that it is now possible to get detailed information about the underlying reaction mechanisms. The data is of high quality, the analysis looks sound, and the conclusions are supported by measurements and interpretations based on available potential energy surfaces.

This work is of significance to the field of molecular reaction dynamics. As the authors mention correctly, these reactions occur in for instance the atmosphere and accurate experimental results for these reactions could be important for developing accurate models of processes occurring in these environments.

Some suggestions and remarks:

- Sometimes, the authors refer to the different product channels using (1a)-(1e), whereas in other instances they write the atomic products formed. For readability, I would recommend to be consistent here.
- As written on page 6, $E_{K_f} = E_K + E_{c.m.} + \Delta E_{v,J} - E_{c.m.} = 0.1$ eV (at maximum), and the values for E_K are given in eq. 1. If I now for instance look at channel 1b in fig. 1, I would expect that, for $v=0, J=0$, $E_{K_f} = 5.45 + 0.1 = 5.55$ eV. However, the blue vertical bar for $v=0$ in fig. 1 is around $E_{K_f} = 5.8$ eV. I would suggest to explain the assignments a bit more in the Supp. Mat..
- Looking at fig. 1(a), it seems to me as if there is another small peak around 1.3 eV. In case this is a real peak, could the authors explain where this one comes from?
- In ref. 8, the Dalitz plot has been simulated. Would it be possible to show these simulations for this system as well? It would be even more convincing than now if that would show similar lines/triangles as the experiment in fig. 2a (ii-iv).
- Table 2 seems to contradict what has been written about the $3s\sigma_g$ state at the beginning of the discussion. It there seemed like the contribution of this intermediate state to the $O(^3P)+O(^3P)$ channel would be very minor. However, that is still 32%. So maybe this could be clarified a bit better by the authors in the text.
- I am wondering why the authors chose to use fig. 4 instead of just adding the fits to fig. 2b. Or, maybe another option if the figure becomes too busy, the three panels of fig. 4 could be placed below one another, with the grey bars for the two different intermediate electronic states, such that one can still see the shifts of the peaks. This could substitute fig. 2b. I feel like the data would be better and more clearly presented in that way, without showing the same data twice.

Some minor points:

- For fig. 2a, it would be helpful to mention in the caption what the red dashed lines in (i) are (mentioned in the text, but not in the caption) and what the arrows indicate in (ii), (iii) and (iv).
- I would recommend to already show in fig. 2b that (1d) has been multiplied by a factor 10. It is mentioned in the caption, but it would be good to also show it in the figure itself.
- On page 4, line 183, I would suggest to use AB^+ instead of O_2^+ , if that is indeed what the authors want to say with this sentence.
- On page 14, line 642, a reference is missing for the sentence about a recent experimental study.
- In the Supp. Mat. on page 2, line 58, "(see, e.g. (see, e.g. [6,8])" should be "(see, e.g. [6,8])".
- In the Supp. Mat. on page 2, line 60 and 65, the black circles must be blue circles, and grey circles must be red circles.
- In the Supp. Mat. on page 5, line 221, it is written that figure S2(b) shows the fit. Maybe I am wrong, but I do not see a fit in this figure. Could it be that this one is missing? Or could it be that the authors would here like to refer to fig. 1 of the main manuscript?

Reviewer #3

(Remarks to the Author)

The paper describes an experimental investigation of mutual neutralization (NM) of O_2^+ and O^- using the so-called DESIREE facility.

Unraveling the NM physics of $O_2^+ + O^-$ is indeed interesting both fundamentally and in several environments and of interest to specialist in more research fields. The detailed analysis and discussion of the fragmentation dynamics in the paper (Figure 2-4) is very interesting and well-explained. Under these aspects I can recommend the paper for publication.

However, I do find the abstract and especially the introduction to be somewhat poorly written both in terms of English and in terms of logical organization of the material, even down to the build up of individual sentences. I find it strange that the authors did not work much more carefully on this text before submitting to Nature Communications. Beyond problems with writing clear sentences, as it stands now, the introductory text interleaves subjects of NM fundamental physics, experiment, application in plasmas and in the atmosphere, and other experiments ($NO^+ + O^-$) in a way that makes the subject appear confusing. This is a shame, as the paper really contains very nice results. I will not go into details of this but simply state that I cannot recommend the paper for publication in any journal if this introduction is not carefully revised and restructured.

One thing that is presently emphasized in the introduction is that the purpose of the paper is to demonstrate the utility of the DESIREE facility for studies of NM of positive molecules and negative atoms. I believe this has already been demonstrated in Ref. 8 and partly also Ref. 21, so I don't see an argument for publishing the methodology in Nature communication. But I am fine with focusing on the NM physics of O_2^+ and O^- .

Specific comments

Section 2.1, Line 248-249: From comparisons of 2-body and 3-body data with one particle removed (Fig. S1), the authors conclude that the 2-body channel is absent. It is sticking, however, that there is actually a difference between the two spectra around 0.5-0.7 eV. I realized that this is a small contribution, and the statement that the NM process is dominated by 3-body channels is true, but I believe the authors should point out a possible small contribution from 2-body channels. Also in Fig. S1 it appears natural to plot the residual normalized to the 2-body value.

Figure 1 and associated text on page 6-7

The peaks in kinetic energy corresponding the 3-body channels are very broad and it is really not so clear that a fit can clarify the vibrational distribution in a meaningful way and even less clear that the peak corresponding to channel (1d) can be fitted with individual vibrational components. It also seems that the temperature of 3000K for the distribution of vibrations leading to channel (1b) and (1d) was simply chosen as a number. I am sure that the temperature must be part of the fitted parameters and should be stated with an errorbar. Thus, I need to see an account of the experimental energy resolution applied in the fit, how the fits were more explicitly made, and a discussion on how meaningful these fits are, and finally also an errorbar on the fitted temperature.

In line 274-275, I could not understand the argument why channel (1e) can be ruled out. There actually seems to be a small peak at low energy in Fig. 1(a)? This should be explained better and directly following the statement.

Smaller comment

There seem to be a confusion about the typesetting and notations of the paper. For example should figures be referred to as e.g. fig. 1a, fig. 1(a) or fig. 1(a)? Another example is that $E_{c.m.}$ occurs with c.m. in italic in some places and others not. A third example is that $r_{[A-B]}$ is used to name a distance of two atoms, while in Fig. 3 this notation is not used. These examples again points to a somewhat careless review of the manuscript before submission.

Reviewer #4

(Remarks to the Author)

This manuscript presents an interesting study of the mutual neutralization of O_2^+ with O^- anions using the merged beam technique as enabled by the dual ion-beam storage ring apparatus DESIREE. DESIREE has opened up a new era in the study of mutual neutralization phenomena, providing information on MN cross sections and the dynamics of MN processes through measurement of the momenta of the neutral products using time- and position-sensitive detection techniques. The experiment also makes use of the cryogenic environment of DESIREE to examine vibrational state dependence for the O_2^+ reactant by examining the reaction dynamics over a wide range of trapping times for $^{16}O^{18}O^+$ ions that can radiatively cool owing to the small dipole moment in this mixed isotopolog.

The O_3 system that is accessed in these experiments is chemically very interesting – at lower levels of excitation O_3 of course forms the reactive ozone molecule of broad atmospheric relevance, but in the ionosphere and other oxygen plasma environments the MN process undoubtedly occurs, probing very high levels of excitation in O_3 . The experimental results are very interesting, and indicate that MN at low relative collision energies leads to three body breakup to form 3O atoms distributed among $O(3P)$ and $O(1D)$ products. By varying the O_2^+ ion internal temperature from 3000 to 1000K, only minimal effects on the branching between $O(3P)$ and $O(1D)$ was observed, however, the kinetic energy release spectra and the details of the momentum partitioning between the three atomic products as measured with Dalitz maps provides strong evidence for the involvement of at least two O_2 Rydberg states in distinct sequential dissociation mechanisms wherein electron transfer from O^- initiates the reaction, with a sequential dissociation of the excited O_2 intermediate. The kinetic energy release spectra similarly show a strong temperature effect on the highest energy product channel observed producing $O(1D) + O(1D) + O(3P)$.

Overall I find this to be a very strong manuscript, with a clear explanation of the convincing data that was collected. I think the

observation of the clear signature of vibrational energy dependence on the accessed states of $O_2^* + O$ provides important information on the role MN of these fundamental molecular and atomic ions can play in a variety of environments, and I therefore support publication in the journal.

I do have one significant question for the authors, however:

The manuscript reports that all MN reactions lead to 3 atomic products by examining the 2-body spectrum and comparing to '2-out-of-3 body data', where a random particle is removed from each 3-body event and then analyzed as a 2-body event. In the supplementary information it notes that this is done on data that 'satisfies center-of-mass filtering'. My question pertains to the center-of-mass filtering. Given the relatively large interaction volume, is the center-of-mass filtering not a strong constraint? I would think that for events with large momenta in all three atoms (such as the inverted triangle in the Dalitz plot for channel 1c shown in Figure 2 of the manuscript) that center-of-mass filtering might be very effective. But, based on Figure S1, this might not be the case. Another way of asking this question might be the effective mass resolution in the two and three-body dissociation cases. Elements of this question are addressed in the text after eq. S6 in the supplementary information, but it might be valuable for the authors comment to comment further on this.

Typographical

1. abstract line 38 '...or O2 discharges.'
2. P.3 second paragraph – sentence from lines 106 to 113. I suggest breaking this up into two sentences as opposed to one long one with a nearly 3-line parenthetical clause.

Version 1:

Reviewer comments:

Reviewer #1

(Remarks to the Author)

I spent sufficient time reading the author's response and the revised manuscript. I was satisfied with the appropriate responses to my comments. I noticed that suggestions and criticisms from other reviewers were taken seriously. The revised manuscript is much improved, more readable, and comprehensive.

My main concern in the previous manuscript, the comparative evaluation with previously published results on MN reactions of simple molecular ions using DESIREE, was resolved by the detailed description of specific O_2^+ and O^- MN interactions in the introduction, which reviewed the importance of O_2^+ and O^- MN interactions. I understood the manuscript to emphasize the first vibration-dependent dynamics, although it is a pity that the vibrational structure is not well distinguished. I also learned that $16O16O^+$ undergoes vibrational cooling due to collisional quenching in the atmosphere.

I hereby consent to this paper being accepted for publication in Nature Comm.

Reviewer #2

(Remarks to the Author)

I would like to thank the authors for their replies, additional explanations, and adjustments made to the manuscript. All of my suggestions have been adequately addressed.

Reviewer #3

(Remarks to the Author)

The authors have addressed the concerns raised by me in a satisfactory way and I can recommend the paper for publication

Reviewer #4

(Remarks to the Author)

I appreciate the revisions made by the authors and think the manuscript is much improved. The introduction now focuses more clearly on the physics revealed by this study of mutual neutralization in the important 'O3' system. The results show the importance of two-step mechanisms for MN through Rydberg states of O_2 , and also reveal convincing evidence that vibrational excitation in O_2^+ mediates enhanced production of two O singlet D atoms, justifying the title 'disentangling vibrationally-dependent molecular dynamics...'.
'

Extensive revisions were made in response to the queries of other reviewers. I am satisfied by the response to my query about center-of-mass gating, and I believe the revised, new, and re-ordering of figures described in detail by the authors has been effective. In particular, careful exposition of the insights to three-body dynamics provided by the Dalitz representations is always a challenge, and I believe these matters are clarified in the revised manuscript.

I believe this work has met the standard needed for publication in Nature Communications.

After some general comments in which we detail several major changes to the text (the main manuscript, the supplementary information) and the figures, we address each of the reviewers in turn. For each point, we provide a response in blue text and the changes that were made in either the main manuscript or the SI. Particular changes to the manuscript are indicated in red text.

General comments:

A particular criticism raised was that we should highlight and focus more on the MN reaction itself, and less on the DESIREE facility. To address this, we have rewritten both the abstract and the introduction, where we now bring sharper and initial focus to the $O_2^+ + O^-$ MN reaction before highlighting the facility to the general reader. The new abstract now reads:

“Product distributions and dynamics of mutual neutralisation (MN) reactions involving even simple molecular ions are largely unknown. Combining cryogenically stored-and-merged ion beams with coincident product-imaging techniques addresses this issue. Application to the reaction of O_2^+ with O^- , which occurs in atmospheric phenomena such as sprites and in high-pressure air or oxygen discharges, reveals that the low-collision-energy MN results predominantly in dissociation of the O_2 molecule. The reaction yields both $O(^3P)$ (84%) and $O(^1D)$ (16%) products, but no $O(^1S)$ products. Analysis of the break-up dynamics of the reaction reveals the dominance of two-step mechanisms involving the $3p\lambda_u$ and $3s\sigma_g$ Rydberg states of O_2 . Use of the $^{16,18}O_2^+$ isotopologue shows that the observed products strongly depend on the vibrational levels of the O_2^+ ion for the channel leading to two $O(^1D)$ products. This type of information has not been available before, and should significantly advance theoretical models of MN processes involving small molecules.”

To bring the $\text{O}_2^+ + \text{O}^-$ MN reaction into focus, we have re-ordered and rewritten the original seven paragraph introduction. We keep the first paragraph essentially the same, where we have changed only “...chemical conversion,” to “...chemical rearrangements,”. The $\text{O}_2^+ + \text{O}^-$ MN reaction is now discussed in the second and third paragraphs, followed by a discussion of MN reactions in general and the DESIREE facility in particular, and we end the Introduction with a presentation of our results. We have completely removed the original third paragraph. The new introduction now reads:

“The evolution of plasma environments is defined and governed by intricate balances between ionizing processes, chemical rearrangements, and neutralisation reactions such as mutual neutralisation (MN) and dissociative recombination (DR). Measuring and explaining these processes in detail is fundamental to understanding and modelling non-local thermal equilibrium (non-LTE) environments, from low-temperature interstellar media [1–3] through cool atmospheric plasma [4–8] and stellar atmospheres [9–11], to high-pressure discharges and high-temperature plasma [12, 13].

The $\text{O}_2^+ + \text{O}^-$ mutual neutralisation reaction, for which we report the most detailed MN study so far, is expected to occur in essentially any high-pressure air or oxygen discharges [14]. Both ions are present in the mesosphere, and, although O^- is nominally a minor constituent, the $\text{O}_2^+ + \text{O}^-$ MN reaction is expected to be of significance, given the large cross sections of MN reactions. For example, recent ionospheric models [15], suggest that the related reaction of O^+ with O^- [7, 16] is a significant source of the $\text{O}(^5\text{S}^\circ) \rightarrow \text{O}(^3\text{P})$ 135.6 nm airglow emission, as observed by the Hopkins Ultraviolet Telescope [17]. Another natural phenomenon where the MN of O_2^+ with O^- is expected to be important, is sprites, (see e.g., [18, 19]) which are upwards discharges occurring in conjunction with lightning. The primary sprite activity occurs for only a

few tens of microseconds but certain features persist for much longer. Because they are transient in a non-LTE environment, the full atmospheric ionic reaction scheme does not have time to develop and the charged-particle density can reach values as high as $\approx 10^{11} \text{ m}^{-3}$ [19]. Therefore, MN, including MN between O_2^+ and O^- , is expected to be important in such environments.

In the MN of O_2^+ with O^- at very low collision energies in the center-of-mass frame ($E_{\text{c.m.}} \approx 0 \text{ eV}$), the following final sets of products are energetically allowed:

where the quoted energies correspond to the kinetic energy released in each reaction outcome, given that both reactants are in their electronic and rovibrational ground states. The quoted values are determined using the NIST atomic spectra database [20] and the NIST Chemistry WebBook [21]. For $E_{\text{c.m.}} \approx 0 \text{ eV}$, reaction (1a) represents all possible final states of pure MN with no molecular dissociation while reactions (1b-1e) represent every possible final channel for MN accompanied by molecular dissociation.

Until recently, experimental studies of MN involving molecular ions in flow tubes [22–24] and merged-beams experiments [25, 26] were limited to measurements of overall reactivities without detailed information on the reaction mechanisms or the final states formed. The Double ElectroStatic Ion Ring ExpEriment (DESIREE)

facility [27, 28], with its combination of stored and merged ion beams and coincident imaging detection, has now made such studies possible [8, 29]. The properties of MN reaction products, including their kinetic energies are of great importance in oxygen-containing plasmas. There, products with large kinetic energies may excite other plasma constituents in collisions or contribute to plasma cooling when leaving the region. In addition, electronically excited atomic and molecular products may instead emit characteristic red (630 nm), green (558 nm) [30, 31], and blue photons [32] responsible for atmospheric airglows. Thus, a good understanding of the properties of atomic and molecular ions and the details of their interactions in natural plasma, including mutual neutralization between O_2^+ with O^- , will help to better understand atmospheric phenomena.

A recent study of the MN of NO^+ and O^- at DESIREE reported that MN is predominantly accompanied by dissociation of the NO molecule, which contrasts to earlier flowing afterglow results for the MN between NO^+ and a number of anions [33–35]. In addition, it was concluded that the reaction occurred via a two-step process involving a pre-dissociating NO^* Rydberg state [8]. However, very little is known about the role of the rovibrational energy in molecular ions on the MN dynamics. To study this effect, long time ion-beam storage is extremely useful, as it allows the ions to relax vibrationally. The reaction can be studied at both short and long storage times and the results be compared. In the related DR process between a free electron and O_2^+ , a strong dependence is observed on the vibrational quanta, with the quantum yields of different oxygen product states varying up to a factor of 2 as a function of vibrational quantum number [30, 31, 36, 37]. Using DESIREE’s unique capabilities, we are now able to investigate the influence of vibrational excitations in a molecular MN reaction for the first time.

Here, we report that the MN of O_2^+ with O^- at low collision energies, $E_{c.m.} \leq 0.1$ eV, is completely dominated by dissociation into three products, in which the process proceeds in a two-step mechanism via Rydberg states in O_2 , which we identify. In addition, we find that the intensity of one of the reaction channels depends strongly on the vibrational state of the O_2^+ parent ion.”

To address several other questions, comments and suggestions from the reviewers, in the main manuscript we have updated Figure 1, Figure 2, and Figure 3, and have removed Figure 4. In the SI we have updated Figure S1, added a new Figure S2, and updated the, now renumbered, Figure S3.

The new Figure 1 and caption is shown here.

Fig. 1: Coincident three-body total kinetic energy release, E_{K_f} , and corresponding Total Displacement, TD , distributions. Experimental data are plotted as filled circles with statistical error bars. Individually simulated distributions are plotted as dashed lines, and the solid line represents the total fit, with the positions of the $\text{O}_2^+(v, J_{\text{avg}})$ rovibrational energies represented by the vertical bars. **a** Data for MN of $^{16,16}\text{O}_2^+ + \text{O}^-$ at storage times 0-10 s ($T_v \sim 3000\text{K}$). **b** Data for MN of $^{16,18}\text{O}_2^+ + \text{O}^-$ at storage times 0-5 s (black circles, $T_v \sim 3000\text{K}$) and 30-60 s (red circles, $T_v \sim 1000\text{K}$).

The new Figure 2 and caption is shown here.

Fig. 2: Dalitz representation analysis for the observed three-body product channels with extracted intermediate kinetic energy release E_{K_i} distributions. **a** Dalitz representation analysis showing (i) the relation between Dalitz coordinates η_1, η_2 and break-up geometries, with the red dotted lines indicating the energy axis of each atomic product; (ii), (iii), and (iv): Experimental Dalitz plots for the E_{K_f} in the reaction channels (1b) ($O(^3P)+O(^3P)+O(^3P)$), (1c) ($O(^1D)+O(^3P)+O(^3P)$), and (1d) ($O(^1D)+O(^1D)+O(^3P)$), respectively. The white arrows denote the energy axis of each oxygen atom, and the white dashed lines are shown to highlight the observed structures. **b** Extracted E_{K_i} distributions for the formation of the different intermediate states of O_2^* in the reaction channels (1b) ($O(^3P)+O(^3P)+O(^3P)$), (1c) ($O(^1D)+O(^3P)+O(^3P)$), and (1d) ($O(^1D)+O(^1D)+O(^3P)$), respectively. Individually simulated distributions are plotted as dashed lines, and the solid line represents the total fit.

The new Figure 3 and caption is shown here.

Fig. 3: Potential energy curves relevant to the $\text{O}_2^+ + \text{O}^-$ MN reaction.
a $3s\sigma_g$ Rydberg states in O_2 . **b** $3p\lambda_u$ Rydberg states in O_2 . Data are adapted from van der Zande *et al.* [41] and Morrill *et al.* [42]. The O_2^+ potential has been shifted by the electron affinity (EA) of O to illustrate the kinetic energies E_{K_i} and E_{K_f} in the two-step process for the reaction $\text{O}_2^+(v=2) + \text{O}^- \rightarrow \text{O}_2^*(v=2) + \text{O}(^3\text{P}) + E_{K_i} \rightarrow \text{O}(^1\text{D}) + \text{O}(^3\text{P}) + \text{O}(^3\text{P}) + E_{K_f}$, as an example.

The new Figure S1 and caption is shown here.

Fig. S1: Kinetic energy release, E_{K_f} , distributions for two-body coincidence data. The data indicated by the filled blue circles shows the measured two-body data, and the red filled circles the contribution from three-body data where one neutral product is randomly removed.

The new Figure S2 and caption is shown here.

Fig. S2: Dalitz plots for the three observed final product channels. Plots on the top, middle, and bottom rows correspond to the reaction channels (1b) ($O(^3P)+O(^3P)+O(^3P)$), (1c) ($O(^1D)+O(^3P)+O(^3P)$), and (1d) ($O(^1D)+O(^1D)+O(^3P)$), respectively. Experimental data are shown in the far right-hand column. The plots in the first two columns show results from simulations of the $3p\lambda_u$ and $3s\sigma_g$ intermediate states, with the relative contribution of these to the observed experimental data shown in the third column.

The new Figure S3 and caption is shown here.

Fig. S3: Storage-time dependent vibrational state distributions and coincident three-body total kinetic energy release, E_{K_f} , distributions. **a** Calculated vibrational state distributions in $^{16,18}\text{O}_2^+$ as function of ion storage time with a starting temperature of 3000 K. DESIREE's ≈ 20 K radiation field [51, 52], and $\Delta v > 1$ transitions, are neglected; **b** Coincident three-body total kinetic energy release, E_{K_f} , and Total Displacement, TD , distributions from MN of $^{16,18}\text{O}_2^+ + \text{O}^-$ at storage times 30-60 s.

Reviewer 1:

In this paper, the authors observed the reaction products from the mutual neutralization of O_2^+ and O^- by combining the merged ion beams and coincident product imaging with the cryogenic ion storage ring DESIREE. They disentangled the detailed mechanisms behind the mutual neutralization process and identified specific intermediate states. They found that the reaction predominantly results in complete dissociation into one of three different sets of electronic states of three atomic oxygen products. They also concluded that the reaction occurs through a sequential process.

In the introduction, the motivation of this study and the background, especially connected with “sprites” as a natural phenomenon, were comprehensively described. The dissociation channels were disentangled by taking advantage of the imaging technique, and the involved dynamics were successfully clarified. As for the theoretical approach, the existing results and information were utilized fully. Thus, I thoroughly evaluated their achievement. However, before this paper, the authors have already published an outstanding paper dealing with $NO^+ O^-$ in Phys. Rev. Lett. [ref. 9]. The experimental technique and analysis procedure employed are the same or very similar. I suppose, partly from this point of view, they determined the title of this paper as “Disentangling vibrationally-dependent...”

There are several motivations for the current paper: i) for the first time, a vibrational-energy-dependent mutual neutralisation reaction can be reported and explained; ii) we also for the first time demonstrate that we can disentangle the multiple contributions of the intermediate Rydberg-states to the final observed fully dissociative product channels.

- In the first figure in this paper, namely Fig. 1(a), I found the spectra of $16O16O^+$ in (a) at the storage time of 0-10s. Without a permanent dipole moment, this

molecule has no chance of being vibrationally cooled. Then, they showed the spectra of $^{16}\text{O}^{18}\text{O}^+$ in Fig. 1(b), which should be vibrationally cooled at the storage time of 30-60s. Finally, I found the spectra of $^{16}\text{O}^{18}\text{O}^+$ at the storage time of 30-60s in Fig. S2(b) in Supplementary Information. I feel they should use Fig. S2(b) as the first figure instead of the present Fig.1(a) or at least add Fig. S2(b) to Fig.1 in the main text.

We have moved Fig. S2b to Fig. 1b, such that the effects of vibrational cooling can be seen more clearly for the MN reaction involving the $^{18,16}\text{O}_2^+$ isotopologue. The original Fig. 1b has been moved to the SI as Fig. S3b. The updated figures are shown above in this reply.

- As evidence of vibrationally-dependent radiative cooling, depression of the population of higher vibrational levels is expected, as shown in colored bars in Fig. 1. However, the observed peak structure is “not vibrationally resolved,” and the corresponding change experimentally observed was not so noticeable (although the higher energy tail was suppressed). But, I understand the most impressive observation is the profile in the range of 1.5-2.0 eV, which is attributed to the component of $\text{O}(1\text{D})+\text{O}(1\text{D})+\text{O}(3\text{P})$. They are explained by the characteristic small potential barrier shown in Fig.3(b). Here, from the viewpoint point of atmospheric chemistry, $^{16}\text{O}^{18}\text{O}^+$ must be a minor component, and $^{16}\text{O}^{16}\text{O}^+$ is not influenced by vibrational cooling.

The use of the $^{18,16}\text{O}_2^+$ isotopologue is purely for the purpose of disentangling and identifying the effects of vibrational energy, which were only hinted at in the $^{16,16}\text{O}_2^+$ isotopologue. As noted by the reviewer, $^{16,16}\text{O}_2^+$ has no dipole moment, and clearly cannot cool down radiatively, but most likely cools via collisional quenching in the

atmosphere, as discussed in E. E. Ferguson, *J. Phys. Chem.* **90**, 731–738 (1986). Furthermore, by using the $^{18,16}\text{O}_2^+$ isotopologue we also access MN reaction data at two different vibrational temperatures, and are able to determine the electronic branching and quantum yields at (internal) temperatures of relevance to, e.g., sprites. Although the present measurements are not vibrationally resolved, the effects of vibrational cooling are clearly observed, and we are able to explain these effects by the small exit barrier on the intermediate $\text{O}_2\ ^1\Delta_u$ potential-energy curve.

Judging from these advantages and drawbacks, I consider that, without a doubt, the presented results are excellent; however, I hesitate to conclude that they are worthy of acceptance in *Nat. Comm.* I'd like to get a response from the authors and other reviewers' opinions.

Finally, I list one question in Supplementary Information SI P.5 line 217 and line 221: I found two different explanations for Fig. S2 (b). I guess the former is the experimental data, and the latter is the simulation result. I think the former is correct.

We apologise for the erroneous description of Fig. S2b. In addressing this reviewer's earlier comment, with updating Fig. 1b to show the data originally plotted in Fig. S2b, we then updated Fig. S3b to show the fit to the late storage-time data of $^{18,16}\text{O}_2^+$. The updated figures are shown above in this reply.

Reviewer 2:

This manuscript presents an experimental study on the mutual neutralisation reaction between O_2^+ and O^- at low collision energy, using DESIREE in combination with coincident product imaging. It was found that at low collision energy, this reaction proceeds predominantly via a two-step mechanism via two classes of Rydberg states of O_2 , resulting in three atomic products via three different reaction channels. The

occurrence of one of these channels depends on the vibrational temperature of the O_2^+ . The data and observations could be interpreted with the help of available potential energy curves for O_2 , such that new insight is provided into the reaction mechanisms for this mutual neutralisation.

In previous research using similar methods, mutual neutralisation of H_3O^+ with OH^- , and NO^+ with O^- was investigated (refs. 21 and 8, respectively). For NO^+ with O^- , one three-body channel was present and it was found that the reaction occurred via a two-step process involving a pre-dissociating NO^* Rydberg state. Here, multiple three-body channels are open, their branching ratio is determined, and information about the intermediate states involved in the two-step process is obtained. So this is really new compared to what has been done for NO^+ with O^- .

It is challenging to investigate mutual neutralisation reactions involving molecular ions, and this study shows that it is now possible to get detailed information about the underlying reaction mechanisms. The data is of high quality, the analysis looks sound, and the conclusions are supported by measurements and interpretations based on available potential energy surfaces.

This work is of significance to the field of molecular reaction dynamics. As the authors mention correctly, these reactions occur in for instance the atmosphere and accurate experimental results for these reactions could be important for developing accurate models of processes occurring in these environments.

We appreciate these very positive remarks.

Some suggestions and remarks: - Sometimes, the authors refer to the different product channels using (1a)-(1e), whereas in other instances they write the atomic products formed. For readability, I would recommend to be consistent here.

We have rewritten the last part of discussion for readability with this comment in mind. The last paragraph before the Discussion section now reads:

“Given the two-step nature of these reactions, further dynamics can be retrieved, and the reaction can be described as follows:

i.e, momentum conservation allows to determine the kinetic-energy release, E_{K_i} , related to the intermediate first step (see the SI). Here, we assume that the O atom no longer participates in the reaction after the initial electron-transfer step, i.e. no further energy is then exchanged between O and O_2^* . Calculated E_{K_i} data are plotted in Fig. 2b, with the top, middle, and bottom panels corresponding to the intermediate states in O_2 involved in the reactions producing $\text{O}(^3\text{P})+\text{O}(^3\text{P})+\text{O}(^3\text{P})$, $\text{O}(^1\text{D})+\text{O}(^3\text{P})+\text{O}(^3\text{P})$, and $\text{O}(^1\text{D})+\text{O}(^1\text{D})+\text{O}(^3\text{P})$, respectively. In the case of reactions producing $\text{O}(^1\text{D})+\text{O}(^3\text{P})+\text{O}(^3\text{P})$, no definitive selection could be made, and a broad background is present. However, the clear peaks observed correspond directly to the features in the associated Dalitz plot. The peaks are narrower in these figures than in the three-body E_{K_f} data, indicating that the intermediate neutral state possesses a similar equilibrium geometry to the parent cation such that vertical electron capture processes are favoured. For the fully ground-state products, $\text{O}(^3\text{P})+\text{O}(^3\text{P})+\text{O}(^3\text{P})$ (top panel), an intense peak is observed, which reinforces the conclusion that one intermediate state is predominantly involved. However, the broad shoulder at larger E_{K_i} does confirm a contribution from a second intermediate state. In the middle panel, corresponding to production of $\text{O}(^1\text{D})+\text{O}(^3\text{P})+\text{O}(^3\text{P})$, the two distinct peaks, with similar amplitude (green curve, corresponding to Fig. 2a(iii)) similarly confirms the involvement of two different intermediate states. Finally, in

the bottom panel, corresponding to production of $O(^1D)+O(^1D)+O(^3P)$, despite the lower level of statistics, two peaks appear in the gold curve, corresponding to Fig. 2a(iv), at similar positions but with different relative amplitudes. ”

- As written on page 6, $E_{K_f} = E_K + E_{c.m} + \Delta E(v, J)$. $E_{c.m} = 0.1$ eV (at maximum), and the values for E_K are given in eq. 1. If I now for instance look at channel 1b in fig. 1, I would expect that, for $v = 0, J = 0$, $E_{Kf} = 5.45 + 0.1 = 5.55$ eV. However, the blue vertical bar for $v = 0$ in fig. 1 is around $E_{Kf} = 5.8$ eV. I would suggest to explain the assignments a bit more in the Supp. Mat..

We determine an initial rotational and vibrational energy distribution for both isotopologues that is described by $T_{vib} = T_{rot} \sim 3000$ K, and these internal energies eventually manifest as kinetic energy of the atomic fragments. To reflect this, we have placed the vertical bars at $E_{K_f} = E_K + E_{c.m} + E_v + E_{J_{avg}}$, where $E_{J_{avg}}$ is the average rotational energy in the ions, hence the observed shift of the distributions.

To clarify the discussion, we have reformulated the indicated equation on page 6 highlighted by the reviewer, and this part of the sentence now reads:

“(iii) the change in rovibrational internal energy between the reactants and products, $\Delta E(v, J)$, i.e., $E_{K_f} = E_K + E_{c.m.} - \Delta E(v, J)$ ”

where we now have a minus sign in front of $\Delta E(v, J)$, since, assuming the notation $\Delta E(v, J) = E_{f(v,J)} - E_{i(v,J)}$, for a dissociative $O_2^+ + O^-$ reaction, $E_{f(v,J)} = 0$ and the value is negative, and thereby a negative sign must be used for it to be added to the products. Furthermore, in the SI after equation (S6) we have added the following two paragraphs and a new equation:

“This equation is exact, if the distance from the point of interaction, L , to the detector would be known with high precision. However, the reaction can happen at any point in the biased interaction region and the average distance is used to compute E , which limits the energy resolution, and results in a broadening of the peaks. An additional broadening is present due to the internal energy in the parent ions, which is completely converted into kinetic energy in the products as:

$$E_{K_f} = E_K + E_{c.m.} + E_{rot} + E_{vib}, \quad (S7)$$

where E_{rot} is the initial rotational energy, E_{vib} the initial vibrational energy, and $E_{c.m.}$ the collision energy. These three quantities have different effects on the final kinetic energy distribution: The vibrational energy mainly results in a tail to higher E_{K_f} values, whereas the rotational and collision energy distribution result in both a broadening and a shift of the E_{K_f} distributions towards higher values. This shift can be observed in the spectra, as the distributions peak at higher values than the expected values (eqn. (1) in the main text).

In the simulations, the broadening due to the length of the interaction region is taken into account by assuming a uniform distribution of events at $L = 1.78 \pm 0.12$ m, corresponding to the size of the biased region. This broadening scales linearly with E_{K_f} , and ranges between 0.5 and 1.5 eV for the channels considered here. The additional broadening effects are taken into account by simulating a Boltzmann distribution of vibrational and rotational energy, and an assumed collision energy of 0.1 ± 0.05 eV (based on previous measurements at DESIREE). A rovibrational energy distribution of 3000 K was found to best describe the data for the $^{16,16}\text{O}_2^+$ ions, and 1000 K vibrational (3000 K rotational) energy distribution for the $^{18,16}\text{O}_2^+$ ions at later storage times of 30-60 seconds. The vertical bars presented in the fits to symbolise the

positions of the vibrational levels, were shifted by both the average rotational and collision energy to reflect this. The fits are highly sensitive to these parameters and we estimate the uncertainty in the temperatures given to be of ± 500 K, as indicated in the main text. ”

- Looking at fig. 1(a), it seems to me as if there is another small peak around 1.3 eV. In case this is a real peak, could the authors explain where this one comes from?

We have updated fig. 1a to mark this feature with an ‘*’ in the spectrum, and the new Figure is shown above in this document. Under the current experimental conditions, data with such small E_{K_f} correspond to events with small time and position separation, for which the false coincidence rate may become important as: i) the flux of neutral events (due to neutralisation in collisions with the extremely thin residual gas) at the center of the detector is higher due to the rather large number of ions in the beams; ii) the possibility that a single event which triggers multiple pixels in the detector is interpreted as two, separate events, i.e., electronic limitations in the detector, and iii) the maximum spot-size determined from the illuminated pixels. Our interpretation of this feature as arising from false coincidences is further supported by the lack of any similar peak in the $^{16,18}\text{O}_2^+$ spectrum, as these data were obtained with significantly lower cation currents which reduces the likelihood of the first two of the contributions given above.

As such, in the main text in the discussion about the kinetic energy release spectra, we now write:

“Although the kinetic energy released into channel (1e) lies only 0.3 eV lower in energy than that released into channel (1d), and could be attributed to the tiny feature observed at lower E_{K_f} values (labelled with a ‘*’ in Fig. 1a), we show later

that this assignment can be ruled out based on the reaction dynamics. Given the small separation ($TD \sim 2$ cm), we assign this tiny feature to false coincidences (see the SI for more details).”

In the SI in connection with the related discussion on the two-body analysis we now write:

“Examination of the residuals shown under the main plot reveals some fluctuations at low E_{K_f} values. Similarly as for the three-particle data, due to the small separations, we assign this as likely arising from false coincidences due to the high flux of neutrals striking the center of the detector from the ion beams, and the limited ability of the detection system to accurately distinguish one-or-two particle hits at such small separations given the practical limitations on the sizes of the individual spots. However, given the statistical uncertainties, we do not rule out a possible small contribution from two-body channels.”

- In ref. 8, the Dalitz plot has been simulated. Would it be possible to show these simulations for this system as well? It would be even more convincing than now if that would show similar lines/triangles as the experiment in fig. 2a (ii-iv).

We thank the reviewer for this suggestion, and agree that Dalitz plot simulations help convince the reader that the fits are meaningful. The simulations show an excellent with the measured distributions, and the results of this type of analysis are now included in the SI as a new Figure S2, as shown above. To describe the data shown in the new Figure S2, we have added the following paragraph to the SI:

“The fit can then be used to reproduce the experimental Dalitz spectra. Separate simulations are performed for the $3s\sigma_g$ and $3p\lambda_u$ intermediate states, and then combined according to the values obtained from the fit. Figure S2 shows the results of

this analysis. As can be seen, the combined simulated Dalitz plots are very similar to the experimental ones, indicating that the fitting procedure was successful.”

- Table 2 seems to contradict what has been written about the $3s\sigma_g$ state at the beginning of the discussion. It there seemed like the contribution of this intermediate state to the $O(^3P)+O(^3P)$ channel would be very minor. However, that is still 32. So maybe this could be clarified a bit better by the authors in the text.

The reviewer is correct that this is somewhat misleading. The intermediate $3s\sigma_g$ state involved in the $O(^3P)+O(^3P)$ channel, i.e., associated with the less-intense shoulder to larger E_{K_i} in the E_{K_i} spectrum, is more difficult to see in the Dalitz plot due to the dominating feature corresponding to $3p\lambda_u$, though some faint features are seen within the triangle. However, these data are well reproduced by the simulations, the results of which are now shown in the new Figure S2 in the SI. To further inform the reader, and in response to suggestions and questions from the reviewers, we have updated the Dalitz plots shown in Figure 2 in the main manuscript with additional white lines to highlight these features. The new Figures and captions are shown above. In updating these, we also made changes in the manuscript. For example, the initial discussion of the data shown in the Dalitz plots now reads:

“Some additional events are observed inside the dominating triangle of the Dalitz plot in Fig. 2a(ii), although of much lower intensity, possibly indicating a second intermediate state. White dashed lines are shown to highlight these features. Although Fig. 2a(iii) shows different structures to that of Fig. 2a(ii), the features again indicate a two-step process. Here, however, two very distinct sets of lines are clearly seen, whose intersections give rise to the bright knots, meaning that this product channel is definitely reached via two different intermediate states. Finally, though with fewer statistics, structures are also identified in Fig. 2a(iv) (as highlighted by the white

dashed lines) for the production of $O(^1D)+O(^1D)+O(^3P)$ ”.

- I am wondering why the authors chose to use fig. 4 instead of just adding the fits to fig. 2b. Or, maybe another option if the figure becomes too busy, the three panels of fig. 4 could be placed below one another, with the grey bars for the two different intermediate electronic states, such that one can still see the shifts of the peaks. This could substitute fig. 2b. I feel like the data would be better and more clearly presented in that way, without showing the same data twice.

We thank the reviewer for this suggestion, and we agree that the data are better presented in this manner. We have thus removed Fig 4, and have updated and simplified Fig. 2. The new Fig. 2 is shown above in this document.

- Some minor points:

- For fig. 2a, it would be helpful to mention in the caption what the red dashed lines in (i) are (mentioned in the text, but not in the caption) and what the arrows indicate in (ii), (iii) and (iv).

In the process of updating this figure, the caption was edited, and this comment addressed. The caption now reads:

“Dalitz representation analysis for the observed three-body product channels with extracted intermediate kinetic energy release E_{K_i} distributions. a Dalitz representation analysis showing (i) the relation between Dalitz coordinates η_1, η_2 and break-up geometries, with the red dotted lines indicating the energy axis of each atomic product; (ii), (iii), and (iv): Experimental Dalitz plots for the E_{K_f} in the reaction channels (1b) $(O(^3P)+O(^3P)+O(^3P))$, (1c) $(O(^1D)+O(^3P)+O(^3P))$, and (1d) $(O(^1D)+O(^1D)+O(^3P))$, respectively. The white arrows denote the energy

axis of each oxygen atom, and the white dashed lines are shown to highlight the observed structures. **b** Extracted E_{K_i} distributions for the formation of the different intermediate states of O_2^* in the reaction channels (1b) ($O(^3P)+O(^3P)+O(^3P)$), (1c) ($O(^1D)+O(^3P)+O(^3P)$), and (1d) ($O(^1D)+O(^1D)+O(^3P)$), respectively. Individually simulated distributions are plotted as dashed lines, and the solid line represents the total fit.”

- I would recommend to already show in fig. 2b that (1d) has been multiplied by a factor 10. It is mentioned in the caption, but it would be good to also show it in the figure itself.

In addressing other aspects raised by the reviewers, Figure 2 has been updated so that this issue is also resolved: the data in Fig 2b) are plotted with absolute counts.

- On page 4, line 183, I would suggest to use AB+ instead of O2+, if that is indeed what the authors want to say with this sentence.

In addressing several other comments/questions, this paragraph as been removed.

- On page 14, line 642, a reference is missing for the sentence about a recent experimental study.

Thank you for the observation. The reference has been added: Western, C. M., Booth, J.-P., Chatterjee, A. & de Oliveira, N. “Rydberg spectra of singlet metastable states of O_2 ”. *Molecular Physics* **119**, e1741714 (2021), and the sentence reads:

“A recent experimental study has identified radiative transitions from this state to the a $^1\Delta_g$ state [47].”

- In the Supp. Mat. on page 2, line 58, “(see, e.g. (see, e.g. [6,8])” should be “(see, e.g. [6,8])”.

This has been corrected.

- In the Supp. Mat. on page 2, line 60 and 65, the black circles must be blue circles, and grey circles must be red circles.

This has been corrected.

- In the Supp. Mat. on page 5, line 221, it is written that figure S2(b) shows the fit. Maybe I am wrong, but I do not see a fit in this figure. Could it be that this one is missing? Or could it be that the authors would here like to refer to fig. 1 of the main manuscript?

We apologise for the erroneous description of Fig. S2b. As the reviewer points out, these data and the fit indeed are shown in Fig 1b in the main manuscript. In addressing questions/comments raised by the other reviewers, we have updated Fig 1 in the main manuscript to more effectively highlight the effect of vibrational cooling to the reader, and have thus also updated the figure in the SI, now Figure S3, to show the fit to the data obtained from the MN reaction of the $^{18,16}\text{O}_2^+$ isotopologue at later storage times.

Reviewer 3:

The paper describes an experimental investigation of mutual neutralization (NM) of O_2^+ and O^- using the so-called DESIREE facility.

-Unraveling the NM physics of $O_2^+ + O^-$ is indeed interesting both fundamentally and in several environments and of interest to specialist in more research fields. The detailed analysis and discussion of the fragmentation dynamics in the paper (Figure 2-4) is very interesting and well-explained. Under these aspects I can recommend the paper for publication.

- However, I do find the abstract and especially the introduction to be somewhat poorly written both in terms of English and in terms of logical organization of the material, even down to the build up of individual sentences. I find it strange that the authors did not work much more carefully on this text before submitting to Nature Communications. Beyond problems with writing clear sentences, as it stands now, the introductory text interleaves subjects of NM fundamental physics, experiment, application in plasmas and in the atmosphere, and other experiments ($NO++O^-$) in a way that makes the subject appear confusing. This is a shame, as the paper really contains very nice results. I will not go into details of this but simply state that I cannot recommend the paper for publication in any journal if this introduction is not carefully revised and restructured.

-One thing that is presently emphasized in the introduction is that the purpose of the paper is to demonstrate the utility of the DESIREE facility for studies of MN of positive molecules and negative atoms. I believe this has already been demonstrated in Ref. 8 and partly also Ref. 21, so I don't see an argument for publishing the methodology in Nature communication. But I am fine with focusing on the MN physics of O_2^+ and O^- .

We thank the reviewer for their suggestion that we lift up the exciting new physics we have unraveled in the MN of O_2^+ and O^- . To achieve this, and, as we go through in detail in our General Comments at the start of our reply, we have rewritten both the

abstract and the introduction, where we now bring sharper and initial focus to the $O_2^+ + O^-$ MN reaction before highlighting the importance of the facility to the general reader. The new Abstract and Introduction is given in the General Comments.

-Specific comments -Section 2.1, Line248-249: From comparisons of 2-body and 3-body data with one particle removed (Fig. S1), the authors conclude that the 2-body channel is absent. It is sticking, however, that there is actually a difference between the two spectra around 0.5-0.7 eV. I realized that this is a small contribution, and the statement that the NM process is dominated by 3-body channels is true, but I believe the authors should point out a possible small contribution from 2-body channels. Also in Fig. S1 it appears natural to plot the residual normalized to the 2-body value.

Similarly to the small feature observed at low kinetic energy releases in Fig 1a, and, as raised by reviewer 2 - and now indicated with an "*" in the updated Fig. 1a - given the small separations involved, we assign this to be most likely due to false coincidences arising from the high flux of neutrals striking the center of the detector, as well as the ability of the detection system to accurately distinguish one-or-two particle hits at such small separations, i.e., the two-from-three particle data subtraction does not take into consideration the real detector's efficiency to distinguish two-from-three particles at very small separations. However, as suggested, we have updated the plot of the residuals, and added a sentence about the two-body channel having possibly a small contributions, which reads:

"Examination of the residuals shown under the main plot reveals some fluctuations at low E_{K_f} values. Similarly as for the three-particle data, due to the small separations, we assign this as likely arising from false coincidences due to the high flux of neutrals striking the center of the detector from the ion beams, and the limited ability of the

detection system to accurately distinguish one-or-two particle hits at such small separations given the practical limitations on the sizes of the individual spots. However, given the statistical uncertainties, we do not rule out a possible small contribution from two-body channels.”

-Figure 1 and associated text on page 6-7

-The peaks in kinetic energy corresponding the 3-body channels are very broad and it is really not so clear that a fit can clarify the vibrational distribution in a meaningful way and even less clear that the peak corresponding to channel (1d) can be fitted with individual vibrational components. It also seems that the temperature of 3000K for the distribution of vibrations leading to channel (1b) and (1d) was simply chosen as a number. I am sure that the temperature must be part of the fitted parameters and should be stated with an errorbar. Thus, I need to see an account of the experimental energy resolution applied in the fit, how the fits were more explicitly made, and a discussion on how meaningful these fit are, and finally also an errorbar on the fitted temperature.

As noted in our response to a question/comment raised by reviewer 2, we have added additional information to the SI, which now reads:

“This equation is exact, if the distance from the point of interaction, L , to the detector would be known with high precision. However, the reaction can happen at any point in the biased interaction region and the average distance is used to compute E , which limits the energy resolution, and results in a broadening of the peaks. An additional broadening is present due to the internal energy in the parent ions, which is completely converted into kinetic energy in the products as:

$$E_{K_f} = E_K + E_{c.m.} + E_{rot} + E_{vib}, \quad (S7)$$

where E_{rot} is the initial rotational energy, E_{vib} the initial vibrational energy, and $E_{\text{c.m.}}$ the collision energy. These three quantities have different effects on the final kinetic energy distribution: The vibrational energy mainly results in a tail to higher E_{K_f} values, whereas the rotational and collision energy distribution result in both a broadening and a shift of the E_{K_f} distributions towards higher values. This shift can be observed in the spectra, as the distributions peak at higher values than the expected values (eqn. (1) in the main text).

In the simulations, the broadening due to the length of the interaction region is taken into account by assuming a uniform distribution of events at $L = 1.78 \pm 0.12$ m, corresponding to the size of the biased region. This broadening scales linearly with E_{K_f} , and ranges between 0.5 and 1.5 eV for the channels considered here. The additional broadening effects are taken into account by simulating a Boltzmann distribution of vibrational and rotational energy, and an assumed collision energy of 0.1 ± 0.05 eV (based on previous measurements at DESIREE). A rovibrational energy distribution of 3000 K was found to best describe the data for the $^{16,16}\text{O}_2^+$ ions, and 1000 K vibrational (3000 K rotational) energy distribution for the $^{18,16}\text{O}_2^+$ ions at later storage times of 30-60 seconds. The vertical bars presented in the fits to symbolise the positions of the vibrational levels, were shifted by both the average rotational and collision energy to reflect this. The fits are highly sensitive to these parameters and we estimate the uncertainty in the temperatures given to be of ± 500 K, as indicated in the main text. ”

We agree that the individual components in channel (1d) are likely not fitted in such a meaningful way that vibrationally resolved branching can be obtained, which is why we refrain from doing so. The fit serves more to highlight the lack of any contribution from lower vibrational levels to the spectra.

-In line 274-275, I could not understand the argument why channel (1e) can be ruled out. There actually seems to be a small peak at low energy in Fig. 1(a)? This should be explained better and directly following the statement.

As noted in a previous response, we attribute the small peak to false coincidences, and have now explicitly pointed out that in the manuscript and the SI. For the peak to be consistent with channel (1e), it would require that the ions producing these fragments have zero ro-vibrational energy and undergo exclusively 0 eV collisions, something which is not motivated by the data observed in the E_{K_f} spectrum. Furthermore, another argument as to why it is excluded is presented later in the main text when discussing the potential energy curves. There, we motivate that channel (1e) can not be accessed as the only available potentials curves which couple out to this set of products lie very far way in energy from the Rydberg states in O_2 that are involved in the current MN reaction (and are thereby not shown in the figure). We have also supplied an additional reference to support this conclusion, which is a new reference [49]: S. L. Guberman, in “Physics of Ion-Ion and Electron-Ion Collisions” (eds F. Brouillard, & J. W. McGowan), (Springer US, Boston, MA, 1983), pp 167–200. In particular, the paragraph motivating our interpretation now reads:

“The exclusion of the $O(^3P)+O(^1S)$ channel, (1e), can be understood from this model: there are no readily accessible potentials that couple out to this channel from the Rydberg states involved here. Dissociative recombination studies indicate namely that this exit channel is only directly accessible from $O_2^+(v > 9)$ [31, 48], as confirmed by the theoretical calculations by Guberman [49]. It is therefore no surprise that it is not observed in the present experiment.”

- Smaller comment

-There seem to be a confusion about the typesetting and notations of the paper. For example should figures be referred to as e.g. fig. 1a, fig. 1a) or fig. 1(a)? Another example is that $E_{c.m}$ occurs with c.m. in italic in some places and others not. A third example is that $r_{[A-B]}$ is used to name a distance of two atoms, while in Fig. 3 this notation is not used. These examples again points to a somewhat careless review of the manuscript before submission.

We have fixed the typesetting inconsistencies, and have implemented the suggestion for inter-nuclear separation in Figure 3, and the new Figure 3 is shown above.

Reviewer 4:

- This manuscript presents an interesting study of the mutual neutralization of O₂⁺ with O⁻ anions using the merged beam technique as enabled by the dual ion-beam storage ring apparatus DESIREE. DESIREE has opened up a new era in the study of mutual neutralization phenomena, providing information on MN cross sections and the dynamics of MN processes through measurement of the momenta of the neutral products using time- and position-sensitive detection techniques. The experiment also makes use of the cryogenic environment of DESIREE to examine vibrational state dependence for the O₂⁺ reactant by examining the reaction dynamics over a wide range of trapping times for ¹⁶O¹⁸O⁺ ions that can radiatively cool owing to the small dipole moment in this mixed isotopologue.

The O₃ system that is accessed in these experiments is chemically very interesting – at lower levels of excitation O₃ of course forms the reactive ozone molecule of broad atmospheric relevance, but in the ionosphere and other oxygen plasma environments the MN process undoubtedly occurs, probing very high levels of excitation in O₃. The experimental results are very interesting, and indicate that MN at low relative collision energies leads to three body breakup to form 3O atoms distributed

among O(3P) and O(1D) products. By varying the O₂⁺ ion internal temperature from 3000 to 1000K, only minimal effects on the branching between O(3P) and O(1D) was observed, however, the kinetic energy release spectra and the details of the momentum partitioning between the three atomic products as measured with Dalitz maps provides strong evidence for the involvement of at least two O₂ Rydberg states in distinct sequential dissociation mechanisms wherein electron transfer from O⁻ initiates the reaction, with a sequential dissociation of the excited O₂ intermediate. The kinetic energy release spectra similarly show a strong temperature effect on the highest energy product channel observed producing O(1D) + O(1D) + O(3P).

- Overall I find this to be a very strong manuscript, with a clear explanation of the convincing data that was collected. I think the observation of the clear signature of vibrational energy dependence on the accessed states of O₂^{*} + O provides important information on the role MN of these fundamental molecular and atomic ions can play in a variety of environments, and I therefore support publication in the journal.

We appreciate these very positive comments and remarks.

- I do have one significant question for the authors, however: - The manuscript reports that all MN reactions lead to 3 atomic products by examining the 2-body spectrum and comparing to ‘2-out-of-3 body data’, where a random particle is removed from each 3-body event and then analyzed as a 2-body event. In the supplementary information it notes that this is done on data that ‘satisfies center-of-mass filtering’. My question pertains to the center-of-mass filtering. Given the relatively large interaction volume, is the center-of-mass filtering not a strong constraint? I would think that for events with large momenta in all three atoms (such as the inverted triangle in the Dalitz plot for channel 1c shown in Figure 2 of the manuscript) that center-of-mass filtering might be very effective. But, based on Figure S1, this might not be the case.

Another way of asking this question might be the effective mass resolution in the two and three-body dissociation cases. Elements of this question are addressed in the text after eq. S6 in the supplementary information, but it might be valuable for the authors comment to comment further on this.

The purpose of the center-of-mass cut is to ensure that real two-body MN signals that could be present in the data are recovered. The detection efficiency for two particles is significantly higher than that for three particles, as is also discussed in references 8 and 31, and, in the absence of such filtering, the signal would be completely dominated by the 2-out-of-3 body data as every combination of products would contribute. Simulations indicate that the approach taken in using the center-of-mass cut does not bias the interpretation, where a center-of-mass distribution on the detector of 5 mm is typically used, and this is the cut that is also applied in the current analysis.

- Typographical - S1. abstract line 38 ‘... or O2 discharges.’
- S2. P.3 second paragraph – sentence from lines 106 to 113. I suggest breaking this up into two sentences as opposed to one long one with a nearly 3-line parenthetical clause.

In answering reviewer 3’s comments on the abstract and the introduction, these particular suggestions have also been addressed.